# Highly efficient UV/H$_2$O$_2$ technology for the removal of nifedipine antibiotics: Kinetics, co-existing anions and degradation pathways

**Wenping Dong**[1,2]☯, **Chuanxi Yang**[3]☯, **Lingli Zhang**[4], **Qiang Su**[1,2], **Xiaofeng Zou**[1,2], **Wenfeng Xu**[5], **Xingang Gao**[6], **Kang Xie**[7], **Weiliang Wang**[3]*

1 Shandong Academy of Environmental Science Co., Ltd., Jinan, China, 2 Shandong Huankeyuan Environmental Engineering Co., Ltd., Jinan, China, 3 School of Environmental and Municipal Engineering, Qingdao University of Technology, Qingdao, China, 4 Hi-tech Science Park Branch of Weihai Municipal Bureau of Ecological Environment, Weihai, China, 5 Shandong Think-eee Environmental Technology Co., Ltd., Jinan, China, 6 Qingdao Jiaming Measurement and Control Technology Co., Ltd., Qingdao, China, 7 School of Civil Engineering and Architecture, University of Jinan, Jinan, China

☯ These authors contributed equally to this work.
* sdqcsdnu@163.com

**Data Availability Statement:** All relevant data are within the manuscript and its Supporting Information files.

## Abstract

This study investigates the degradation of nifedipine (NIF) by using a novel and highly efficient ultraviolet light combined with hydrogen peroxide (UV/H$_2$O$_2$). The degradation rate and degradation kinetics of NIF first increased and then remained constant as the H$_2$O$_2$ dose increased, and the quasi-percolation threshold was an H$_2$O$_2$ dose of 0.378 mmol/L. An increase in the initial pH and divalent anions (SO$_4^{2-}$ and CO$_3^{2-}$) resulted in a linear decrease of NIF (the R$^2$ of the initial pH, SO$_4^{2-}$ and CO$_3^{2-}$ was 0.6884, 0.9939 and 0.8589, respectively). The effect of monovalent anions was complex; Cl$^-$ and NO$_3^-$ had opposite effects: low Cl$^-$ or high NO$_3^-$ promoted degradation, and high Cl$^-$ or low NO$_3^-$ inhibited the degradation of NIF. The degradation rate and kinetics constant of NIF *via* UV/H$_2$O$_2$ were 99.94% and 1.45569 min$^{-1}$, respectively, and the NIF concentration = 5 mg/L, pH = 7, the H$_2$O$_2$ dose = 0.52 mmol/L, T = 20 ˚C and the reaction time = 5 min. The ·OH was the primary key reactive oxygen species (ROS) and ·O$_2^-$ was the secondary key ROS. There were 11 intermediate products (P345, P329, P329-2, P315, P301, P274, P271, P241, P200, P181 and P158) and 2 degradation pathways (dehydrogenation of NIF → P345 → P274 and dehydration of NIF → P329 → P315).

## 1 Introduction

Water pollution is a major environmental problem the world is facing today, mainly due to modernization [1]. The removal of toxic organic pollutants discharged from the ever-increasing number of industries is a major environmental goal [2,3]. Nifedipine (NIF, Fig 1), 3,5-dimethyl 2,6-dimethyl-4-(2-nitrophenyl)-1,4-dihydropyridine-3,5-dicarboxylate, belongs to the dihydropyridine class of calcium channel antagonists and is one of the most useful pharmaceuticals for the treatment of hypertension, angina pectoris and other cardiovascular

**Funding:** This work was supported by the National Natural Science Foundation of China (41672340) and the Research and Demonstration of Special Reagents for Sewage Treatment Plant in Chemical Industry Park (RD28-2019).

**Competing interests:** The authors have declared that no competing interests exist.

disorders [4,5]. As a large portion of each administered dose is excreted from medical applications and the pharmaceutical industry, and a substantial amount of NIF is released to the environment [6]. It has been demonstrated that NIF residues in the environment can result in the evolution of novel antibiotic-resistant bacteria that ultimately pose a threat to the aquatic ecosystem and human health through human organ lesions and increased bacterial resistance [7,8]. Hence, the efficient removal of NIF from water is significant and essential to reducing environmental and ecological risks.

The removal of antibiotics from aqueous solutions has been widely researched, including removal by physical methods, chemical methods and biological methods [9–12]. Adsorption and advanced oxidation processes (AOPs, such as photocatalysis, Fenton, Fenton-like, photo-Fenton and catalytic ozonation) are the most promising wastewater treatment technologies for the removal of antibiotics from water environments and reduction of the resulting environmental risks because they are fast, efficient, low cost and convenient [13–16]. Many adsorbents have been employed for the eradication of antibiotics [17]. However, there are some drawbacks, such as incomplete removal, high energy requirements and the generation of toxic sludge and other waste products that entail further disposal [18]. Solar light-driven photocatalysis involves the photoinduced generation of holes (h$^+$) in the valence band (VB) and electrons (e$^-$) in the conduction band (CB) *via* light absorption by a semiconductor (TiO₂, ZnO and CdS). Sequential interfacial charge transfers release various reactive oxygen species (ROS), such as superoxide, peroxide, and hydroxyl radicals, which participate in the degradation of organic and inorganic pollutants [19–22]. However, the limitations of a wide bandgap, the rapid recombination rate of photogenerated electron-hole pairs, low solar light energy utilization efficiency, photocorrosion, and poor recyclability reduce the photocatalytic efficiency

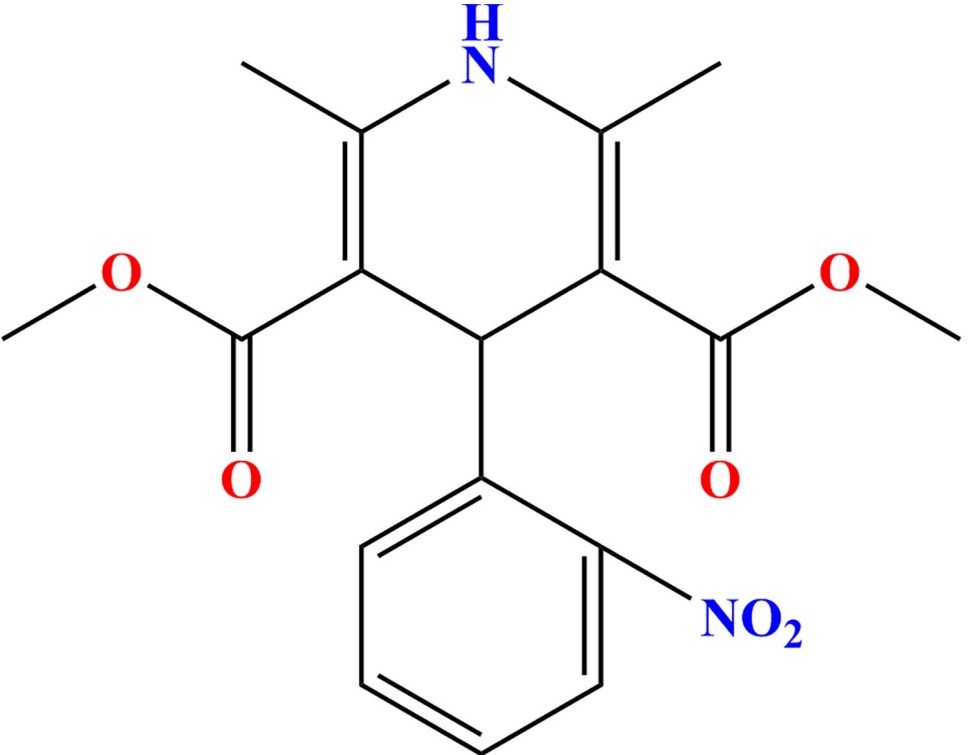

**Fig 1. Structural formula of NIF.**

[23–25]. It is imperative to develop a novel Z-scheme system or heterojunction photocatalyst with broad photocatalytic applications [26,27]. However, limited research has focused on the removal of NIF from the water environment *via* AOPs. Therefore, it is important to study the removal of NIF *via* AOPs for the treatment of medical wastewater.

NIF is a known light-sensitive drug that degrades *via* intramolecular mechanisms to 4-(2-nitrophenyl) pyridine homolog (under UV light irradiation) and 4-(2-nitrosophenyl)-pyridine homolog (under daylight irradiation) [28]. Mojtaba Shamsipur et al. used a multivariate curve resolution method based on the combination of the Kubista approach and an iterative target transformation method by Gemperline to study the kinetics of NIF decomposition upon exposure to a 40 W lamp [29]. The results indicated that the photodecomposition kinetics of NIF are zero-order at the beginning of the reaction. However, when the reaction was more than 50% complete, the kinetics of the reaction changed to a first-order mechanism. The photo-degradation kinetics constants for the zero-order and first-order regions were $(4.96 \pm 0.13)^* 10^{-9}$ $M^{-1}$ $s^{-1}$ and $(6.22 \pm 0.10)^* 10^{-5}$ $s^{-1}$, respectively. This was the first study on the degradation of NIF, but the low degradation rate (65%) and kinetics limited the application of NIF removal *via* a photo-degradation system.

A novel method of UV light combined with hydrogen peroxide (UV/H₂O₂) is highly efficient, fast, and has a strong oxidizing ability; these advantages are attributed to the synergistic ability of UV light and H₂O₂ to generate ROS [30]. However, in UV/H₂O₂ AOPs, other constituents in water matrices may significantly affect the removal of target contaminants by competitively interacting with photons and ROS. In our previous study, the degradation of norfloxacin by using UV/H₂O₂ was investigated [31]. The degradation rate and apparent first-order kinetics constant of norfloxacin *via* UV/H₂O₂ were 98.8% and 0.22248 $min^{-1}$, respectively, and the norfloxacin concentration = 20 mg/L, the H₂O₂ dose = 1.2 mmol/L, the pH = 7, T = 20°C and the reaction time = 20 min. The kinetics were low, and the formation mechanism of ROS was controversial, but it provided a novel research direction for the degradation of NIF *via* a UV/H₂O₂ system. Therefore, it should be noted that the degree of research to date on the degradation of NIF *via* UV/H₂O₂ oxidation processes is insufficient to thoroughly understand the fundamentals of ·OH generation, intermediate products and degradation pathways, which are important processes that must be considered in the design of wastewater treatment technology [32]. Furthermore, the effect of co-existing anions in the UV/H₂O₂ system may significantly affect the removal of NIF by competitively quenching with ROS [33]. Thus, it is still challenging to design a UV/H₂O₂ wastewater treatment technology with high efficiency.

On the one hand, the oxidizability of UV/H₂O₂ AOPs and removal rate of NIF were enhanced due to the combination between UV and H₂O₂ [30]. On the other hand, the anion (such as $NO_3^-$) was generated due to the degradation reaction between NIF and ROS. However, impacts of $NO_3^-$ showed duality: it promotes the generation of ROS under irradiation, and also quenches the ROS of UV/H₂O₂ [32,33]. Hence, it is significant and meaningful to study the effect of co-existing anions on the degradation of NIF via UV/H₂O₂ AOPs. To better understand the removal efficacy of a target compound by UV/H₂O₂ AOPs in different real water environments, the divalent anions ($SO_4^{2-}$ and $CO_3^{2-}$) and monovalent anions ($Cl^-$ and $NO_3^-$) have been developed to model the impact of water constituents on the reaction kinetics.

The aims of this study were to demonstrate the application of NIF degradation and to evaluate the performance and mechanism of UV/H₂O₂ AOPs. The specific objectives were (1) to assess the effect of the H₂O₂ dose, initial pH, and co-existing anions ($SO_4^{2-}$, $CO_3^{2-}$, $Cl^-$ and $NO_3^-$) on the degradation of UV/H₂O₂, (2) to predict the key ROS of the UV/H₂O₂ method and (3) to propose the degradation pathway of NIF.

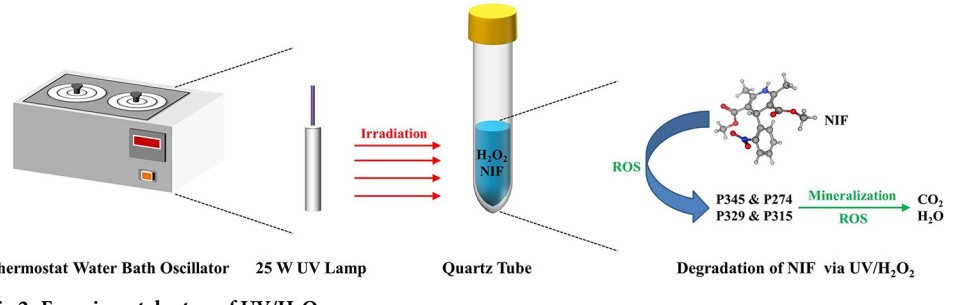

**Fig 2. Experimental setup of UV/H$_2$O$_2$.**

## 2 Materials and methods

### 2.1 Chemicals

NIF was purchased from Shanghai Aladdin Bio-Chem Technology Co., Ltd. (Shanghai, China). Hydrogen peroxide (H$_2$O$_2$), hydrochloric acid (HCl), sodium hydroxide (NaOH), sodium sulfate (Na$_2$SO$_4$), sodium carbonate (Na$_2$CO$_3$), sodium nitrate (NaNO$_3$) and sodium chloride (NaCl) were purchased from Sinopharm Chemical Reagent Co., Ltd. (Shanghai, China). Methyl alcohol (CH$_3$OH) was purchased from Thermo Fisher Scientific (Shanghai, China). All chemicals and reagents used were of analytical grade or higher and directly used without further purification. All solutions were prepared with deionized water.

### 2.2 Experimental setup

The UV/H$_2$O$_2$ degradation experiments (Fig 2) were conducted in deionized water with the addition of H$_2$O$_2$ to the sample prior to 25 W UV light source exposure (254 nm). The initial NIF concentration was 5 mg/L, the temperature was 20˚C, the H$_2$O$_2$ dose was 0–1.04 mmol/L and the pH was 4–10. To understand the effect of co-existing anions, different sources of SO$_4$$^{2-}$, CO$_3$$^{2-}$, Cl$^-$ and NO$_3$$^-$ (from 5 to 50 mg/L) were added to the NIF degradation experiments to evaluate the removal rate and degradation kinetics.

### 2.3 Removal rate and degradation kinetics

The removal rate ($\eta$) of NIF under UV/H$_2$O$_2$ was calculated using Eq 1 (Eq 1):

$$\eta = \frac{C_0 - C_t}{C_0} \times 100\% \qquad \text{(Eq 1)}$$

where $C_0$ is the initial concentration of NIF and $C_t$ is the concentration of NIF at a certain degradation time, which was determined from the liquid chromatogram (S1 and S2 Figs).

The degradation kinetics of NIF *via* UV/H$_2$O$_2$ followed the apparent first-order kinetic law, and the apparent first-order kinetic constant ($k'_{app}$) was described by Eq 2 (Eq 2):

$$-\ln\frac{C_t}{C_0} = k'_{app}t \qquad \text{(Eq 2)}$$

where $t$ is the reaction time.

### 2.4 Organics analysis

NIF and its intermediate products in the UV/H$_2$O$_2$ degradation reaction solutions were analyzed by an Agilent 1260 series liquid chromatogram mass spectrometry (LC-Q-TOF-MS)

system (Agilent, USA) with a C18 column (100 mm × 2.1 mm, 3.5 mm). The wavelength was 237 nm according to ultraviolet and visible spectrophotometry (S1 Fig). The mobile phase was methyl alcohol and deionized water at 63:35 (v/v). The drying gas of $N_2$ was 8.0 mL/min, and the testing time was 30 min.

## 2.5 Electron spin resonance (ESR) measurements

ESR measurements were performed with a JES-FA200 electron spin resonance spectrometer and used to measure the hydroxide radical (·OH) and superoxide radical (·O$_2^-$) during the degradation of NIF under UV/$H_2O_2$ using 5,5-dimethyl-1-pyrroline N-oxide (DMPO) as the spin trapping reagent.

# 3 Results and discussion

## 3.1 Effect of $H_2O_2$ dose

In general, the $H_2O_2$ dose significantly affects the oxidative degradation of antibiotics by controlling the generation rate of ROS, and the effect of $H_2O_2$ dose has been shown to have a dual nature. The specific degradation performance of NIF was enhanced by increasing the dose of $H_2O_2$ when it was low; however, the degradation performance of NIF increased slowly, remained constant or decreased when the $H_2O_2$ dose was high. As shown in Fig 3A and 3B and S1 Table, the degradation rates of NIF under UV/$H_2O_2$ were 72.81% (0 mmol/L) < 95.97% (0.13 mmol/L) < 99.31% (0.26 mmol/L) < 99.94% (0.52 mmol/L) < 99.95% (1.04 mmol/L), the kinetics constants k'$_{app}$ were 0.2560 min$^{-1}$ (0 mmol/L) < 0.6752 min$^{-1}$ (0.13 mmol/L) < 1.03947 min$^{-1}$ (0.26 mmol/L) < 1.45569 min$^{-1}$ (0.52 mmol/L) < 1.59404 min$^{-1}$ (1.04 mmol/L), and the t$_{1/2}$ (time of NIF degradation rate = 50%) were 0.4 min (0.52 mmol/L) < 0.6 min (1.04 mmol/L) < 1.0 min (0.26 mmol/L) < 1.5 min (0.13 mmol/L) < 2.8 min (0 mmol/L) when the NIF concentration = 5 mg/L, the $H_2O_2$ dose = 0–1.04 mmol/L, the pH = 7, T = 20 ˚C and the reaction time = 5 min. As shown in Fig 3C, the effect of the $H_2O_2$ dose on the degradation of NIF *via* UV/$H_2O_2$ system had a dual nature. The degradation kinetics constant noticeably increased as the $H_2O_2$ dose increased and then remained constant at 1.5±0.1 min$^{-1}$. When the $H_2O_2$ dose was < 0.52 mmol/L, the slope was 3.013 (min$^{-1}$)/(mmol/L), but it decreased to 0.266 (min$^{-1}$)/(mmol/L) when the $H_2O_2$ dose was > 0.52 mmol/L; hence, the quasi-percolation threshold (QPT) of the $H_2O_2$ dose was 0.378 mmol/L [34]. This trend was based on the generation and quenching of ·OH described by (Eq 3) to (Eq 6) [35]:

(a) $H_2O_2$ dose < QPT :

$$H_2O_2 \rightarrow 2 \cdot OH \qquad (Eq\ 3)$$

(b) $H_2O_2$ dose > QPT :

$$H_2O_2 + \cdot OH \rightarrow \cdot HO_2 + H_2O \qquad (Eq\ 4)$$

$$\cdot HO_2 + \cdot OH \rightarrow H_2O + O_2 \qquad (Eq\ 5)$$

$$2 \cdot OH \rightarrow H_2O_2 \qquad (Eq\ 6)$$

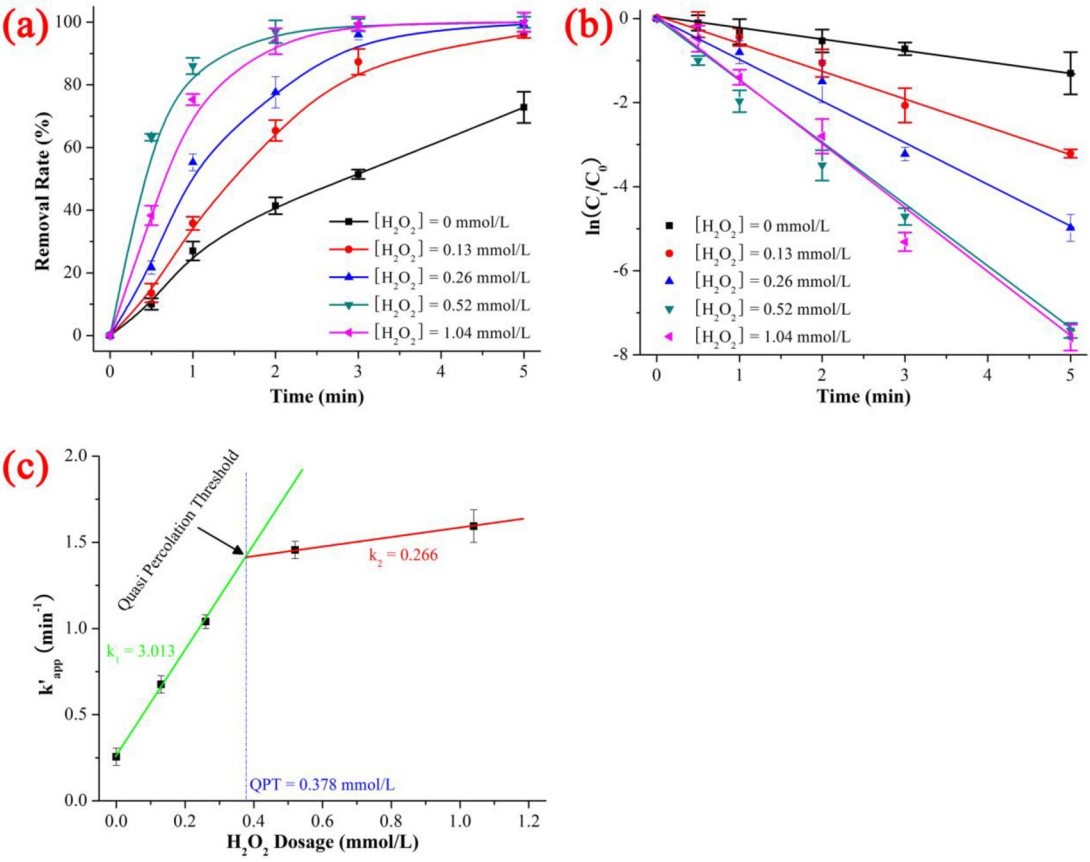

**Fig 3.** Effect of H$_2$O$_2$ dose on removal rate (a), kinetics constant (b) and linear fitting between kinetics constant and H$_2$O$_2$ dose on the degradation of NIF *via* UV/H$_2$O$_2$. The error bars represent the standard deviation (n = 3).

### 3.2 Effect of initial pH

The pH is another key parameter of the UV/H$_2$O$_2$ system. It significantly affects the oxidative degradation of antibiotics by transforming the protonation states and changing the redox potential at different pH values. As shown in Fig 4A and 4B and S2 Table, the degradation rates of NIF under UV/H$_2$O$_2$ was 97.54% (pH = 10) < 98.69% (pH = 8) < 99.77% (pH = 5) < 99.94% (pH = 4 and 7), the kinetics constant k'$_{app}$ was 0.75217 min$^{-1}$ (pH = 10) < 0.91269 min$^{-1}$ (pH = 8) < 1.21831 min$^{-1}$ (pH = 5) < 1.45569 min$^{-1}$ (pH = 7) < 1.51175 min$^{-1}$ (pH = 4), and the t$_{1/2}$ was 0.4 min (pH = 7) < 0.7 min (pH = 4) < 0.8 min (pH = 5) < 0.9 min (pH = 8) < 1.1 min (pH = 10) when the NIF concentration = 5 mg/L, pH = 4–10, H$_2$O$_2$ dose = 0.52 mmol/L, T = 20 °C and reaction time = 5 min. As shown in Fig 4C, the degradation kinetics of NIF *via* UV/H$_2$O$_2$ system exhibited a poor linear decrease as the pH increased (y = -0.1155x + 1.95533, R$^2$ = 0.6884). The results indicated that acidic solution (pH = 4) was more favorable for degrading NIF than basic solution (pH = 10) under UV/H$_2$O$_2$. The possible reasons were in accord with the redox potential, generation rate of ROS and reaction rate between ROS and NIF. The inhibiting effect of the basic solution was due to the quenching reaction between OH$^-$ and ·OH, which is shown in (Eq 7) to (Eq 11) [36]:

$$H_2O_2 \rightarrow HO_2^- + H^+ \qquad (Eq\ 7)$$

$$H^+ + OH^- \rightarrow H_2O \qquad (Eq\ 8)$$

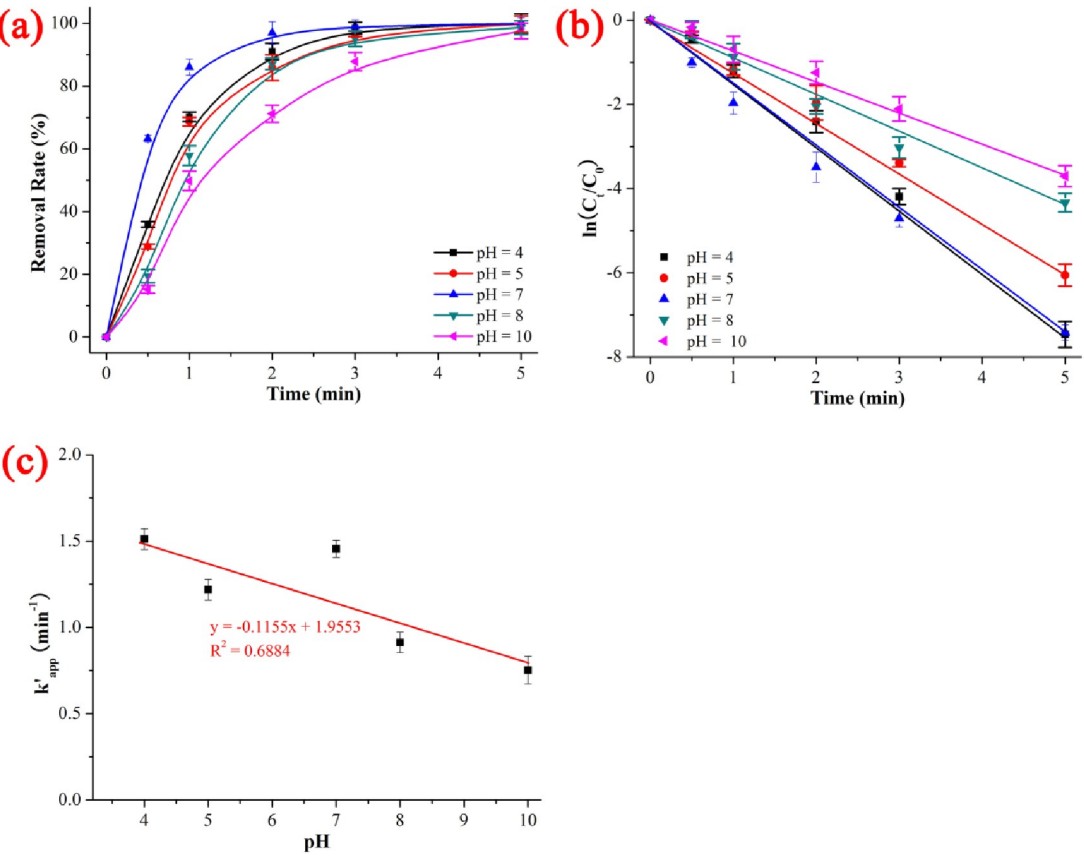

**Fig 4.** Effect of initial pH on the removal rate (a), kinetics constant (b) and linear fitting between kinetics constant and initial pH on the degradation of NIF *via* UV/H$_2$O$_2$. The error bars represent the standard deviation (n = 3).

$$H_2O_2 + HO_2^- \rightarrow H_2O + OH^- + O_2 \qquad \text{(Eq 9)}$$

$$\cdot OH + HO_2^- \rightarrow OH^- + \cdot HO_2 \qquad \text{(Eq 10)}$$

$$\cdot OH + HO_2^- \rightarrow H_2O + \cdot O_2^- \qquad \text{(Eq 11)}$$

## 3.3 Effect of SO$_4$$^{2-}$

It is important to evaluate the effect of co-existing anions (such as SO$_4$$^{2-}$, CO$_3$$^{2-}$, Cl$^-$ and NO$_3$$^-$) because the co-existing anions in wastewater impact the degradation capacity and the oxidation mechanism of ROS.

As shown in Fig 5A and 5B and S3 Table, the degradation rates of NIF under UV/H$_2$O$_2$ with different SO$_4$$^{2-}$ concentrations were 99.26% (SO$_4$$^{2-}$ concentration = 50 mg/L) < 99.81% (SO$_4$$^{2-}$ concentration = 20 mg/L) < 99.93% (SO$_4$$^{2-}$ concentration = 5 mg/L) < 99.94% (SO$_4$$^{2-}$ concentration = 0 mg/L), the kinetics constant k'$_{app}$ was 1.00154 min$^{-1}$ (SO$_4$$^{2-}$ concentration = 50 mg/L) < 1.24540 min$^{-1}$ (SO$_4$$^{2-}$ concentration = 20 mg/L) < 1.38175 min$^{-1}$ (SO$_4$$^{2-}$ concentration = 5 mg/L) < 1.45569 min$^{-1}$ (SO$_4$$^{2-}$ concentration = 0 mg/L), and the t$_{1/2}$ was 0.4 min (SO$_4$$^{2-}$ concentration = 0 mg/L and 5 mg/L) < 0.6 min (SO$_4$$^{2-}$ concentration = 20 mg/L) < 0.8 min (SO$_4$$^{2-}$ concentration = 50 mg/L) when the NIF concentration = 5 mg/L, SO$_4$$^{2-}$

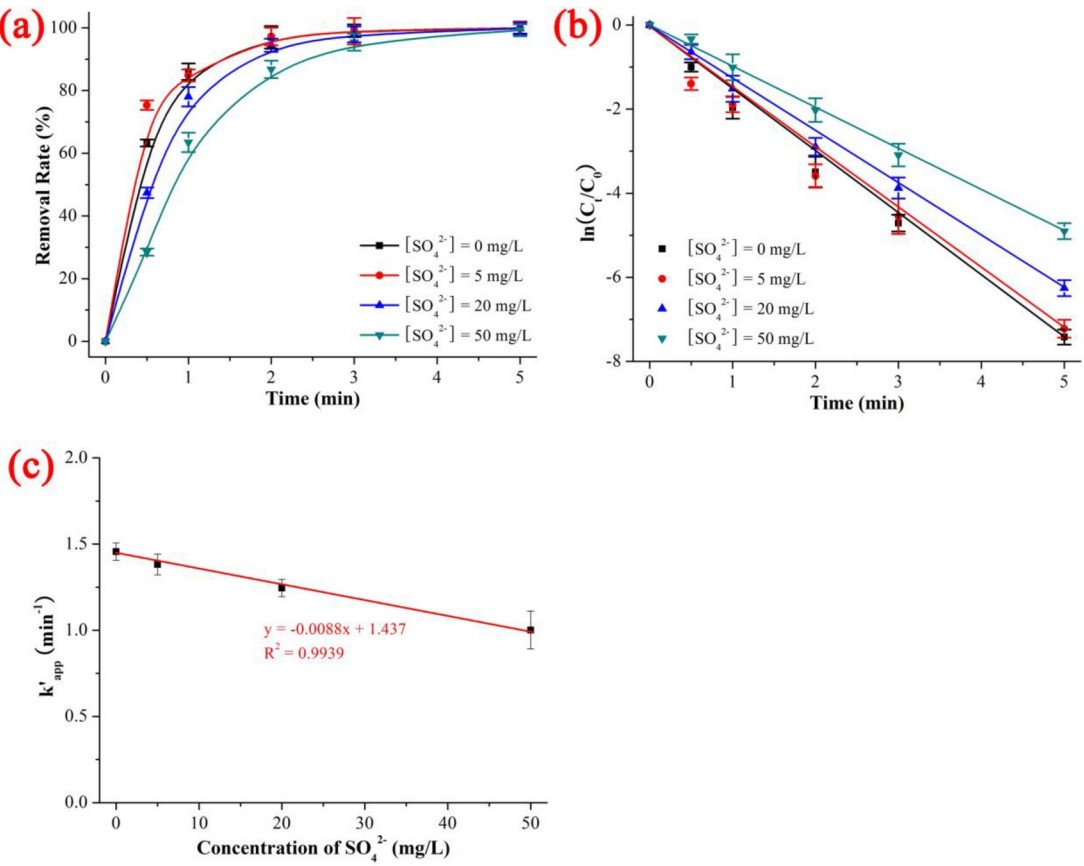

**Fig 5.** Effect of $SO_4^{2-}$ on the removal rate (a), kinetics constant (b) and linear fitting between kinetics constant and concentration of $SO_4^{2-}$ on the degradation of NIF *via* UV/H₂O₂. The error bars represent the standard deviation (n = 3).

concentration = 0–50 mg/L, pH = 7, H₂O₂ dose = 0.52 mmol/L, T = 20 ˚C and reaction time = 5 min. As shown in Fig 5C, the degradation kinetics of NIF *via* UV/H₂O₂ system exhibited a good linear decrease as the $SO_4^{2-}$ concentration increased (y = -0/0088x + 1.437, $R^2$ = 0.9939). The inhibition effect was 5.08% ($SO_4^{2-}$ concentration = 5 mg/L) < 14.45% ($SO_4^{2-}$ concentration = 20 mg/L) < 31.20% ($SO_4^{2-}$ concentration = 50 mg/L), which was in keeping with the trend of the kinetics constants: 1.00154 $min^{-1}$ ($SO_4^{2-}$ concentration = 50 mg/L) < 1.24540 $min^{-1}$ ($SO_4^{2-}$ concentration = 20 mg/L) < 1.38175 $min^{-1}$ ($SO_4^{2-}$ concentration = 5 mg/L). The degradation rate of NIF decreased with increasing $SO_4^{2-}$ concentration, and the quenching mechanism of ROS *via* $SO_4^{2-}$ is shown in (Eq 12) to (Eq 17) [37]:

$$H^+ + SO_4^{2-} \rightarrow HSO_4^- \tag{Eq 12}$$

$$HSO_4^- + \cdot OH \rightarrow SO_4 \cdot^- + H_2O \tag{Eq 13}$$

$$SO_4 \cdot^- + H_2O \rightarrow H^+ + SO_4^{2-} + \cdot OH \tag{Eq 14}$$

$$SO_4 \cdot^- + H_2O_2 \rightarrow H^+ + SO_4^{2-} + H_2O \tag{Eq 15}$$

$$SO_4 \cdot^- + H_2O \cdot \rightarrow H^+ + SO_4^{2-} + O_2 \tag{Eq 16}$$

$$SO_4 \cdot^- + e^- \to SO_4{}^{2-} \tag{Eq 17}$$

## 3.4 Effect of CO$_3$$^{2-}$

As shown in Fig 6A and 6B and S4 Table, the degradation rates of NIF under UV/H$_2$O$_2$ with different CO$_3$$^{2-}$ concentrations was 99.41% (CO$_3$$^{2-}$ concentration = 50 mg/L) < 99.71% (CO$_3$$^{2-}$ concentration = 20 mg/L) < 99.85% (CO$_3$$^{2-}$ concentration = 5 mg/L) < 99.94% (CO$_3$$^{2-}$ concentration = 0 mg/L), the kinetics constant k'$_{app}$ was 1.04447 min$^{-1}$ (CO$_3$$^{2-}$ concentration = 50 mg/L) < 1.16907 min$^{-1}$ (CO$_3$$^{2-}$ concentration = 20 mg/L) < 1.29550 min$^{-1}$ (CO$_3$$^{2-}$ concentration = 5 mg/L) < 1.45569 min$^{-1}$ (CO$_3$$^{2-}$ concentration = 0 mg/L), and the t$_{1/2}$ was 0.4 min (CO$_3$$^{2-}$ concentration = 0 mg/L) < 0.5 min (CO$_3$$^{2-}$ concentration = 5 mg/L) < 0.6 min (CO$_3$$^{2-}$ concentration = 20 mg/L) < 0.7 min (CO$_3$$^{2-}$ concentration = 50 mg/L) when the NIF concentration = 5 mg/L, CO$_3$$^{2-}$ concentration = 0–50 mg/L, pH = 7, H$_2$O$_2$ dose = 0.52 mmol/L, T = 20 °C and reaction time = 5 min. As shown in Fig 6C, the degradation kinetics of NIF *via* the UV/H$_2$O$_2$ system exhibited a good linear decrease as the CO$_3$$^{2-}$ concentration increased (y = -0.0072x + 1.3771, R$^2$ = 0.8589). The inhibition trend was 11.00% (CO$_3$$^{2-}$ concentration = 5 mg/L) < 19.69% (CO$_3$$^{2-}$ concentration = 20 mg/L) < 28.25% (CO$_3$$^{2-}$ concentration = 50 mg/L), which was in keeping with that of kinetics constant: 1.04447 min$^{-1}$ (CO$_3$$^{2-}$ concentration = 50 mg/L) < 1.16907 min$^{-1}$ (CO$_3$$^{2-}$ concentration = 20 mg/L) < 1.29550 min$^{-1}$ (CO$_3$$^{2-}$ concentration = 5 mg/L). The degradation rate of NIF decreased with increasing CO$_3$$^{2-}$ concentration, and the quenching mechanism of the ROS *via* CO$_3$$^{2-}$ is shown in (Eq 18) to (Eq 22) [38]:

$$CO_3{}^{2-} + H^+ \to HCO_3{}^- \tag{Eq 18}$$

$$HCO_3{}^- \to CO_3{}^{2-} + H^+ \tag{Eq 19}$$

$$CO_3{}^{2-} + \cdot OH \to CO_3 \cdot^- + OH^- \tag{Eq 20}$$

$$HCO_3{}^- + \cdot OH \to CO_3 \cdot^- + H_2O \tag{Eq 21}$$

$$CO_3 \cdot^- + H_2O_2 \to HO_2 \cdot + HCO_3{}^- \tag{Eq 22}$$

## 3.5 Effect of Cl$^-$

As shown in Fig 7A and 7B and S5 Table, the degradation rates of NIF under UV/H$_2$O$_2$ with different Cl$^-$ concentrations was 99.57% (Cl$^-$ concentration = 50 mg/L) < 99.94% (Cl$^-$ concentration = 0 mg/L) < 99.98% (Cl$^-$ concentration = 20 mg/L) < 100% (Cl$^-$ concentration = 5 mg/L), the kinetics constant k'$_{app}$ was 1.09588 min$^{-1}$ (Cl$^-$ concentration = 50 mg/L) < 1.45569 min$^{-1}$ (Cl$^-$ concentration = 0 mg/L) < 1.72666 min$^{-1}$ (Cl$^-$ concentration = 20 mg/L) < 1.98350 min$^{-1}$ (Cl$^-$ concentration = 5 mg/L), and the t$_{1/2}$ was 0.3 min (Cl$^-$ concentration = 5 mg/L and 20 mg/L) < 0.4 min (Cl$^-$ concentration = 0 mg/L) < 0.9 min (Cl$^-$ concentration = 50 mg/L) when the NIF concentration = 5 mg/L, Cl$^-$ concentration = 0–50 mg/L, pH = 7, H$_2$O$_2$ dosage = 0.52 mmol/L, T = 20 °C and reaction time = 5 min. As shown in Fig 7C, although the degradation kinetics of NIF *via* UV/H$_2$O$_2$ system decreased with increasing Cl$^-$ concentration, the effect of Cl$^-$ on the degradation of NIF had a dual nature: low Cl$^-$ concentrations promoted the degradation of NIF, while high Cl$^-$ concentrations inhibited the degradation of NIF. The degradation kinetics of NIF *via* UV/H$_2$O$_2$ system exhibited a poor linear decrease as the Cl$^-$ concentration increased (y = -0.012x + 1.7897, R$^2$ = 0.5013). The trend of inhibition was

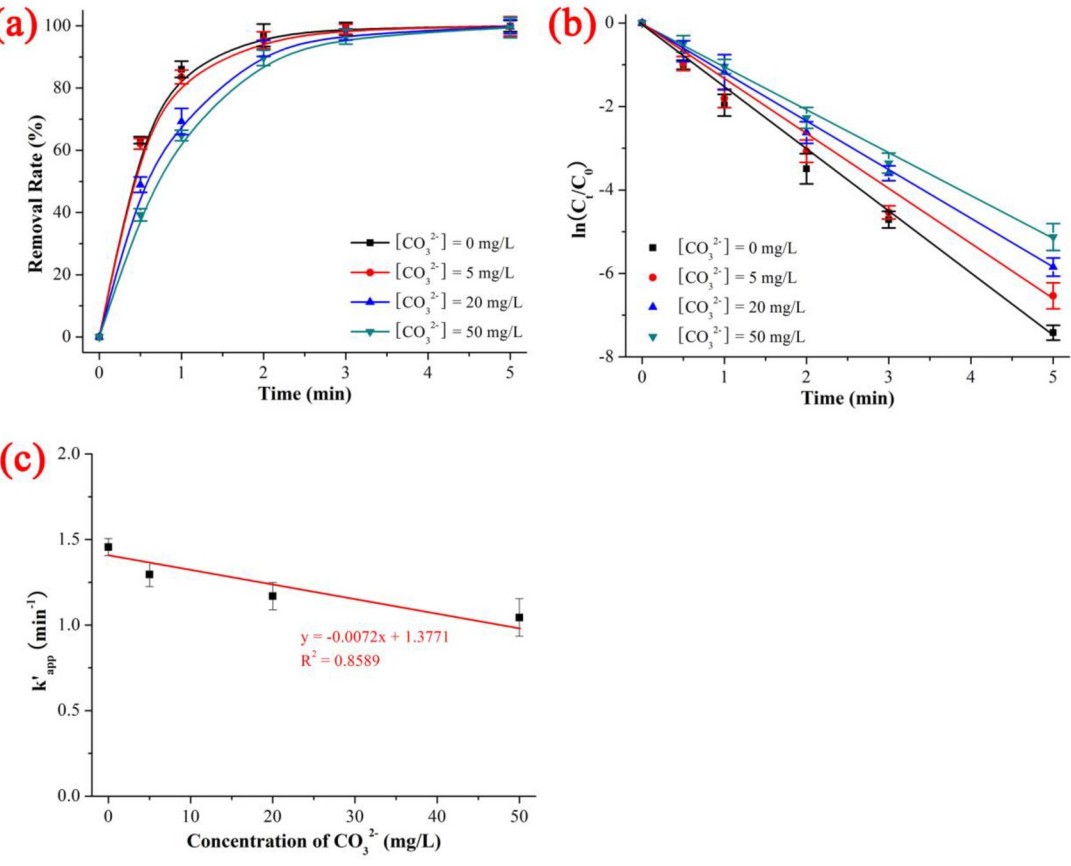

**Fig 6.** Effect of CO$_3^{2-}$ on the removal rate (a), kinetics constant (b) and linear fitting between kinetics constant and concentration of CO$_3^{2-}$ on the degradation of NIF *via* UV/H$_2$O$_2$. The error bars represent the standard deviation (n = 3).

-36.25% (Cl$^-$ concentration = 5 mg/L) < -18.61% (Cl$^-$ concentration = 20 mg/L) < 24.71% (Cl$^-$ concentration = 50 mg/L), which was in keeping with the trend of kinetics constant: 1.09588 min$^{-1}$ (Cl$^-$ concentration = 50 mg/L) < 1.72666 min$^{-1}$ (Cl$^-$ concentration = 20 mg/L) < 1.98350 min$^{-1}$ (Cl$^-$ concentration = 5 mg/L). The reaction mechanism between Cl$^-$ and ·OH is shown in (Eq 23) to (Eq 27) [39]:

$$Cl^- + \cdot OH \rightarrow Cl \cdot + OH^- \tag{Eq 23}$$

$$Cl \cdot + Cl^- \rightarrow \cdot Cl_2^- \tag{Eq 24}$$

$$\cdot Cl_2^- + H_2O_2 \rightarrow H^+ + 2Cl^- + H_2O \cdot \tag{Eq 25}$$

$$\cdot OH + Cl^- \rightarrow \cdot HOCl^- \tag{Eq 26}$$

$$\cdot HOCl^- + H^+ \rightarrow H_2O + Cl^- \tag{Eq 27}$$

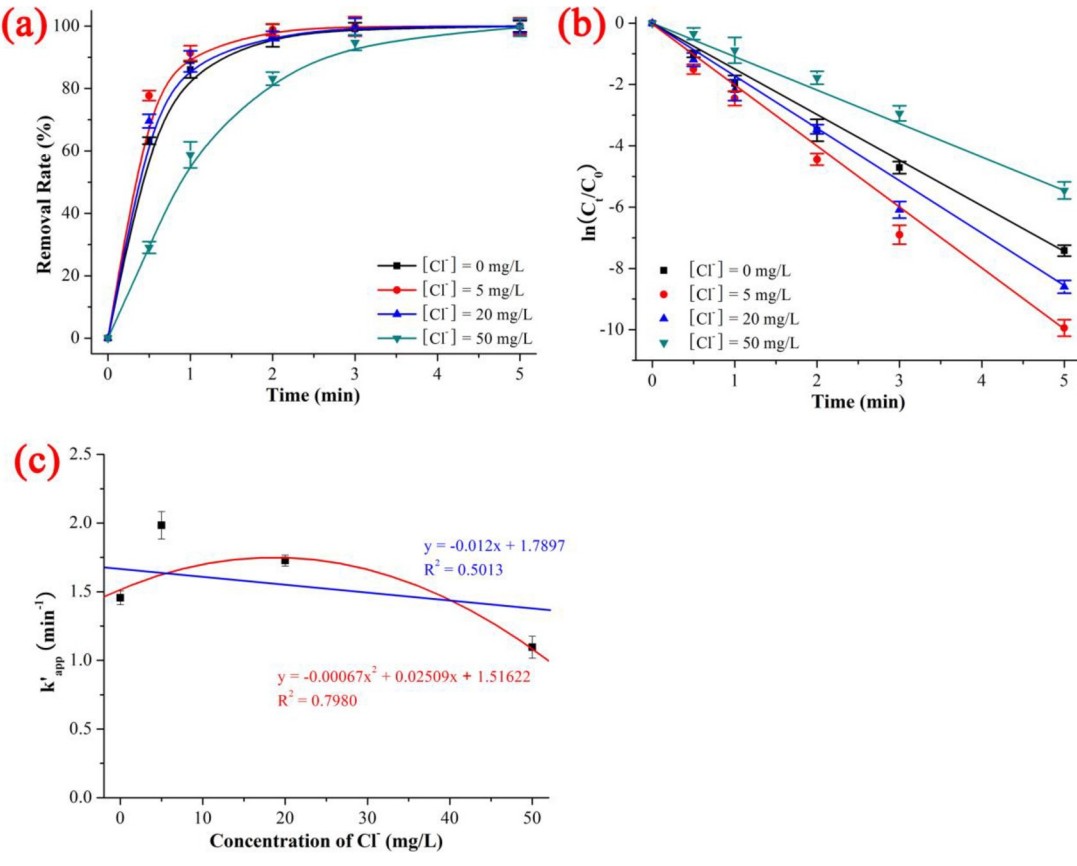

**Fig 7.** Effect of CO$_3^{2-}$ on the removal rate (a), kinetics constant (b) and linear fitting between kinetics constant and concentration of CO$_3^{2-}$ on the degradation of NIF *via* UV/H$_2$O$_2$. The error bars represent the standard deviation (n = 3).

## 3.6 Effect of NO$_3^-$

As shown in Fig 8A and 8B and S6 Table, the degradation rates of NIF under UV/H$_2$O$_2$ with different NO$_3^-$ concentrations was 99.45% (NO$_3^-$ concentration = 5 mg/L) < 99.89% (NO$_3^-$ concentration = 20 mg/L) < 99.94% (NO$_3^-$ concentration = 0 mg/L) < 99.97% (NO$_3^-$ concentration = 50 mg/L), the kinetics constant k'$_{app}$ was 1.03215 min$^{-1}$ (NO$_3^-$ concentration = 5 mg/L) < 1.29801 min$^{-1}$ (NO$_3^-$ concentration = 20 mg/L) < 1.45569 min$^{-1}$ (NO$_3^-$ concentration = 0 mg/L) < 1.55295 min$^{-1}$ (NO$_3^-$ concentration = 50 mg/L), and the t$_{1/2}$ was 0.3 min (NO$_3^-$ concentration 50 mg/L) < 0.4 min (NO$_3^-$ concentration = 0 mg/L and 20 mg/L) < 0.6 min (NO$_3^-$ concentration = 5 mg/L) when the NIF concentration = 5 mg/L, NO$_3^-$ concentration = 0–50 mg/L, pH = 7, H$_2$O$_2$ dose = 0.52 mmol/L, T = 20°C and reaction time = 5 min. As shown in Fig 8C, NO$_3^-$ had the opposite effect on the degradation of NIF *via* the UV/H$_2$O$_2$ system compared to Cl$^-$. The effect of NO$_3^-$ on the degradation of NIF had a dual nature: low NO$_3^-$ concentrations inhibited the degradation of NIF, but high NO$_3^-$ concentrations promoted the degradation of NIF. The degradation kinetics of NIF *via* the UV/H$_2$O$_2$ system showed a poor linear decrease with increasing NO$_3^-$ concentration (y = -0.005377x + 1.31881, R$^2$ = 0.4514). The inhibition effect was -6.68% (NO$_3^-$ concentration = 50 mg/L) < 10.83% (NO$_3^-$ concentration = 20 mg/L) < 29.10% (NO$_3^-$ concentration = 5 mg/L), which was in keeping with the trend of the kinetics constant: 1.03215 min$^{-1}$ (NO$_3^-$ concentration = 5 mg/L) < 1.29801 min$^{-1}$ (NO$_3^-$ concentration = 20 mg/L) < 1.55295 min$^{-1}$ (NO$_3^-$ concentration = 50 mg/L). The mechanism of the reaction between

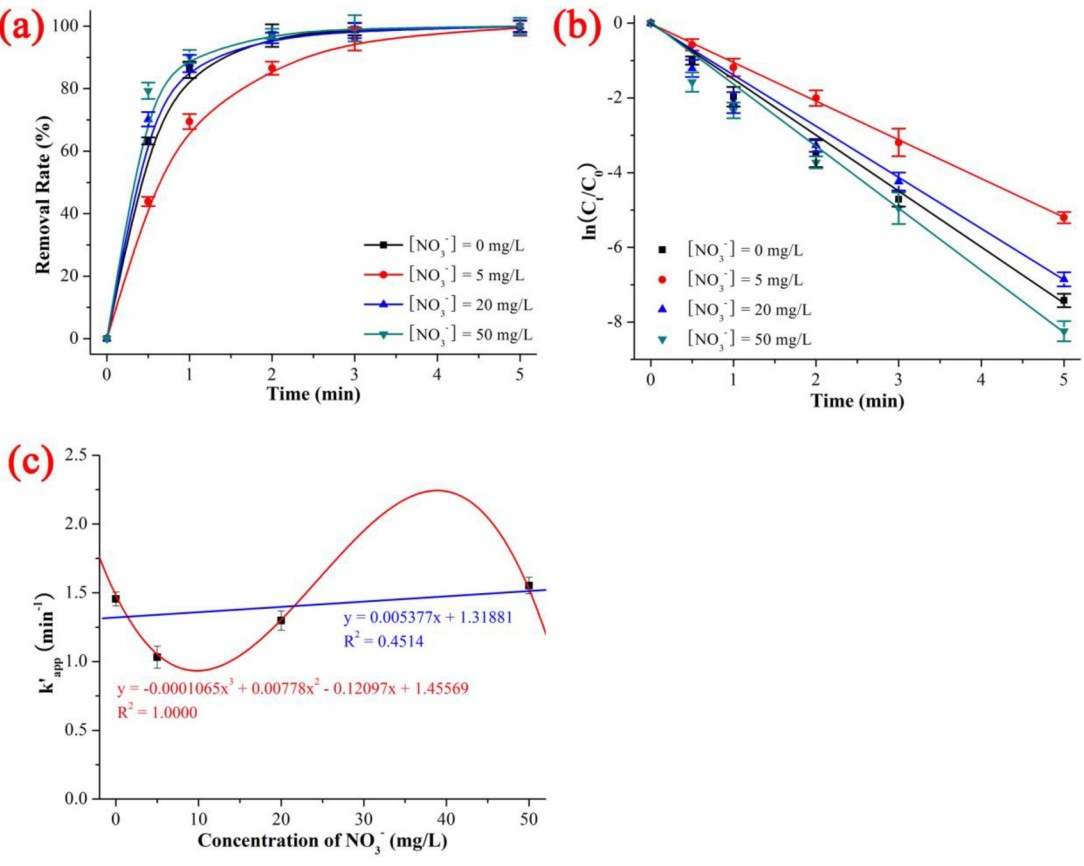

**Fig 8.** Effect of $NO_3^-$ on the removal rate (a), kinetics constant (b) and linear fitting between kinetics constant and concentration of $NO_3^-$ on the degradation of NIF *via* UV/H₂O₂. The error bars represent the standard deviation (n = 3).

$NO_3^-$ and ·OH is shown in (Eq 28) to (Eq 32) [40]:

$$NO_3^- + h\nu \rightarrow NO_2^- + O \qquad \text{(Eq 28)}$$

$$NO_3^- + h\nu \rightarrow NO_2 \cdot + O \cdot^- \qquad \text{(Eq 29)}$$

$$2\,NO_2 \cdot + H_2O \rightarrow NO_2^- + NO - + 2\,H^+ \qquad \text{(Eq 30)}$$

$$O + H_2O \rightarrow 2 \cdot OH \qquad \text{(Eq 31)}$$

$$O \cdot^- + H_2O \rightarrow \cdot OH + OH^- \qquad \text{(Eq 32)}$$

In summary, the effect of co-existing anions was as follows: The divalent anions ($SO_4^{2-}$ and $CO_3^{2-}$) caused a good linear decrease with increasing initial pH and concentration of divalent anions ($R^2$ of $SO_4^{2-}$ and $CO_3^{2-}$ was 0.9939 and 0.8589, respectively). The monovalent anions had a complex effect; $Cl^-$ and $NO_3^-$ had opposite effects on NIF degradation: low $Cl^-$ and high $NO_3^-$ promoted degradation, while high $Cl^-$ and low $NO_3^-$ inhibited degradation. The degradation kinetics of NIF *via* the UV/H₂O₂ system showed a poor linear decrease with increasing $Cl^-/NO_3^-$ concentration ($R^2$ of $Cl^-$ and $NO_3^-$ was 0.5013 and 0.4514, respectively).

## 3.7 Oxidation mechanism and degradation pathway of NIF *via* UV/H$_2$O$_2$

In recent advances in UV/H$_2$O$_2$ systems, the degradation of organic pollutants has been due to the generation of ROS, especially ·OH and ·O$_2^-$. The oxidation mechanism of NIF degradation *via* the UV/H$_2$O$_2$ system was measured by ESR measurements, and the ESR spectra are shown in Fig 9A and 9B. The significant ·OH signal (Fig 9A) showed four peaks at 321.8 mT ($P_A$), 323.3 mT ($P_B$), 324.8 mT ($P_C$) and 326.3 mT ($P_D$). The interspaces of $P_A$-$P_B$, $P_B$-$P_C$ and $P_C$-$P_D$ were constant of 1.5 mT, and the intensity ratio of $P_A$, $P_B$, $P_C$ and $P_D$ was 1:2:2:1 [41]. The significant ·O$_2^-$ signal (Fig 9B) showed four peaks at 322.1 mT ($P'_A$), 323.2 mT ($P'_B$), 324.4 mT ($P'_C$) and 325.8 mT ($P_D$). The interspaces of $P'_A$-$P'_B$, $P'_B$-$P'_C$ and $P'_C$-$P'_D$ were constant within the range of 1.0 mT from 1.5 mT, and the intensity ratio of $P'_A$, $P'_B$, $P'_C$ and $P'_D$ was 1:1:1:1 [42]. However, the intensity of the ·OH signal was stronger than that of the ·O$_2^-$ signal, which means that ·OH was the primary key ROS and ·O$_2^-$ was the secondary key ROS. The oxidation mechanism of NIF degradation *via* the UV/H$_2$O$_2$ system is shown in (Eq 33) to (Eq 41) [43]. The generation of ·OH mainly comes from the direct decomposition of H$_2$O$_2$ (Eq 33), the generation of ·O$_2^-$ mainly comes from the indirect reaction between oxygen gas (dissolved oxygen and H$_2$O$_2$ decomposition) and electrons (Eqs 35 and 38), and NIF is degraded *via* the ROS (Eq 41) [44].

$$H_2O_2 + h\nu \rightarrow 2 \cdot OH \tag{Eq 33}$$

$$H_2O_2 + \cdot OH \rightarrow \cdot HO_2 + H_2O \tag{Eq 34}$$

$$\cdot HO_2 \rightarrow \cdot O_2^- + H^+ \tag{Eq 35}$$

$$2 \cdot HO_2 \rightarrow H_2O_2 + O_2 \tag{Eq 36}$$

$$\cdot HO_2 + \cdot OH \rightarrow H_2O + O_2 \tag{Eq 37}$$

$$O_2 + e^- \rightarrow \cdot O_2^- \tag{Eq 38}$$

$$2 \cdot O_2^- \rightarrow {}^1O_2 + {}^3O_2 + 2e^- \tag{Eq 39}$$

$$2 \cdot OH \rightarrow H_2O_2 \tag{Eq 40}$$

$$NIF + ROS \rightarrow Intermediate\ Products \rightarrow CO_2 + H_2O \tag{Eq 41}$$

The identification of intermediate NIF products *via* UV/H$_2$O$_2$ was performed using an Agilent 1260 series liquid chromatogram mass spectrometry (LC-Q-TOF-MS) system. The reaction conditions were: NIF concentration = 5 mg/L, pH = 7, H$_2$O$_2$ dose = 0.52 mmol/L, T = 20°C and reaction time = 30 min. Preliminary analyses were conducted to evaluate the intermediate products of NIF produced by the UV/H$_2$O$_2$ method, and four kinds of NIF degradation intermediate products were observed in the mass spectrum based on the mass-to-charge (*m/z*) ratio. Four intermediate products were present: P345 (*m/z* = 345), P329 (*m/z* = 329), P315 (*m/z* = 315) and P274 (*m/z* = 274). The structures of P345, P329, P315 and P274 are shown in S3A–S3D Fig, respectively.

As shown in Fig 10A and 10B, the peak areas of all the observed products first increased and then decreased within 30 min. The peak areas of P345 and P274 at different reaction times

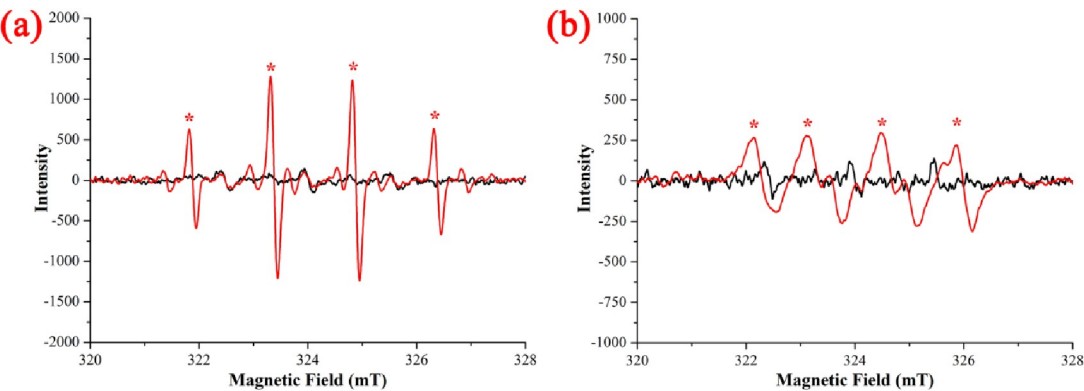

**Fig 9.** ESR measurements of ·OH (a) and ·O₂⁻ (b).

are shown in Fig 10A, and the results indicated that the maximum peak area of P345 appeared at 5 min, the maximum peak area of P274 appeared at 20 min, and the peak intensity of P345 was stronger than that of P274. The first degradation pathway of NIF *via* the UV/H₂O₂ method was proposed as follows: protonated NIF ($C_{17}H_{19}N_2O_6^+$, *m/z* = 347) first lost $H_2$ to generate P345 ($C_{17}H_{17}N_2O_6^+$, *m/z* = 345) *via* dehydrogenation reaction, and then P345 lost $C_3H_5NO$ to generate P274 ($C_{14}H_{12}NO_5^+$, *m/z* = 274), as shown in Fig 11A [45,46]. The peak areas of P329 and P315 at different reaction times are shown in Fig 10B, and the results indicated that the maximum peak area of P329 appeared at 0.5 min, the maximum peak area of P315 appeared at 7 min, and the peak intensity of P329 was stronger than that of P315. The second degradation pathway of NIF *via* the UV/H₂O₂ method was proposed as follows: protonated NIF ($C_{17}H_{19}N_2O_6^+$, *m/z* = 347) first lost $H_2O$ to generate P329 ($C_{17}H_{17}N_2O_5^+$, *m/z* = 329) *via* dehydration reaction, and then P329 lost $CH_2$ to generate P315 ($C_{16}H_{15}N_2O_5^+$, *m/z* = 274), as shown in Fig 11B [45,46].

P315 was degraded with the attack of ROS. The intermediate products of P271 (*m/z* = 271), P241 (*m/z* = 241) and P181 (*m/z* = 181) are shown in S7 Table, and its possible degradation pathway is shown in S4A Fig. P315 first lost $CO_2$ to generate P271 ($C_{15}H_{15}N_2O_3^+$, *m/z* = 271), P271 lost $CH_2O$ to generate P241 ($C_{14}H_{13}N_2O_2^+$, *m/z* = 241), and finally, P241 lost $C_2H_4O_2$ to generate P181 ($C_{12}H_9N_2^+$, *m/z* = 181) [46]. P329 was present as an isomeride that was named P329-2 (*m/z* = 329). The intermediate products, P329-2, P301 (*m/z* = 301), P200 (*m/z* = 200)

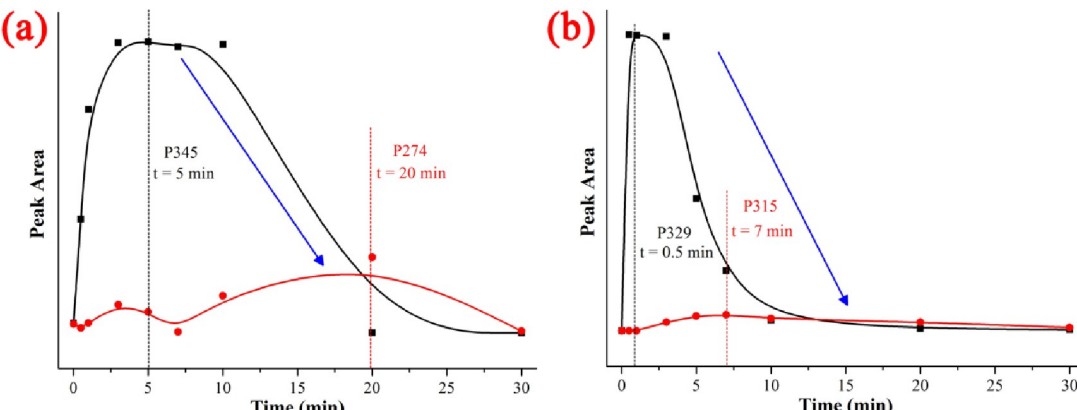

**Fig 10. Chromatography of the intermediate products of NIF *via* UV/H₂O₂.**

**Fig 11. Degradation pathway of NIF *via* UV/H$_2$O$_2$ system.**

and P158 ($m/z$ = 158), are shown in S7 Table, and their possible degradation pathways are shown in S4B Fig. P329-2 first lost CO to generate P301 ($C_{16}H_{17}N_2O_4^+$, $m/z$ = 301), P301 lost $C_4H_7NO_2$ to generate P200 ($C_{12}H_{10}NO_2^+$, $m/z$ = 200), and finally, P200 lost $C_2H_2O$ to generate P158 ($C_{10}H_8NO^+$, $m/z$ = 158) [46].

**Table 1. Summary of removal performance of antibiotics *via* AOPs.**

| Technology | Pollutant | Removal Rate | $k'_{app}$ | Time | Reference |
|---|---|---|---|---|---|
| UV/H₂O₂ | NIF | 99.94% | 1.45569 min$^{-1}$ | 5 min | This work |
| Photo-degradation | NIF | 65% | $(6.22\pm0.1)*10^{-5}$ s$^{-1}$ | 300 min | [29] |
| UV/H₂O₂ | Norfloxacin | 98.8% | 0.22248 min$^{-1}$ | 20 min | [31] |
| Photocatalysis | Norfloxacin | 97% | | 60 min | [49] |
| Photocatalysis | Norfloxacin | 91% | 0.02279 min$^{-1}$ | 90 min | [50] |
| Photocatalysis | Ciprofloxacin | 92.3% | 0.0438 min$^{-1}$ | 50 min | [51] |
| Photo-Fenton | Norfloxacin | 90% | 0.076 min$^{-1}$ | 120 min | [52] |
| O₃ | Norfloxacin | >90% | 0.1935 min$^{-1}$ | 15 min | [53] |
| MnOₓ/SBA-15/O₃ | Norfloxacin | >90% | 0.3147 min$^{-1}$ | 15 min | [53] |
| Sonocatalysis | Norfloxacin | 69.07% | 0.0075 min$^{-1}$ | 150 min | [54] |

## 3.8 Environmental significance

In this paper, fast, effective and low-cost UV/H₂O₂ was used in the degradation of the antibiotic NIF, and this work contributed to the sustainable development of new methods for applications in hospital and aquaculture wastewater treatment for sustainable development, cleaner production and an environmentally friendly society, as shown in Table 1. The maximum degradation rate (99.94%), degradation kinetics constant (1.45569 min$^{-1}$) and minimum degradation time (5 min) indicated that the UV/H₂O₂ system is a promising AOP treatment for organic and medical wastewater. In addition, the cost of the UV/H₂O₂ system was approximately \$0.447 for 1 m$^3$ wastewater (S8 Table), which was lower than that of related systems (ranging from \$0.53 to \$0.85 for 1 m$^3$ wastewater) [47]. During the catalytic oxidation process, all the molecular mechanisms of ROS generation under the UV/H₂O₂ system, the effects of co-existing anions in an actual water environment, the analysis of intermediate products and the degradation pathways were the basis of the efficient AOP design. Furthermore, intermediate products and the degradation pathways of pollutants should also be studied through theoretical simulation technologies such as density functional theory (DFT) and molecular dynamics (MD) [48].

## 4 Conclusions

The degradation rate and degradation kinetics of NIF first increased and then remained constant as the H₂O₂ dose increased, and the quasi-percolation threshold was an H₂O₂ dose of 0.378 mmol/L. The effect of the initial pH, divalent anions ($SO_4^{2-}$ and $CO_3^{2-}$) and monovalent anions ($Cl^-$ and $NO_3^-$) decreased linearly with increasing initial pH and co-existing anions (the $R^2$ values of the initial pH, $SO_4^{2-}$, $CO_3^{2-}$ $Cl^-$ and $NO_3^-$ were 0.6884, 0.9939, 0.8589, 0.5013 and 0.4514, respectively). ·OH was the primary key ROS, and ·$O_2^-$ was the secondary key ROS. There were 11 intermediate products (P345, P329, P329–2, P315, P301, P274, P271, P241, P200, P181 and P158) and 2 degradation pathways (dehydrogenation reaction of NIF → P345 → P274 and dehydration reaction of NIF → P329 → P315).

## Supporting information

**S1 Fig. UV-Vis absorption spectra of NIF.**
(DOCX)

**S2 Fig. Chromatography and standard curve of NIF.**
(DOCX)

**S3 Fig. Mass spectrum of intermediate products of NIF *via* UV/H$_2$O$_2$.**
(DOCX)

**S4 Fig. Possible degradation pathways of P315 and P329-2 in the UV/H$_2$O$_2$ system.**
(DOCX)

**S1 Table. Effect of H$_2$O$_2$ dose on the degradation of NIF *via* UV/H$_2$O$_2$.** Reaction conditions: NIF concentration = 5 mg/L, H$_2$O$_2$ dose = 0–1.04 mmol/L, pH = 7, T = 20 ˚C and reaction time = 5 min.
(DOCX)

**S2 Table. Effect of initial pH on the degradation of NIF via UV/H$_2$O$_2$.**
(DOCX)

**S3 Table. Effect of SO$_4^{2-}$ on the degradation of NIF *via* UV/H$_2$O$_2$.**
(DOCX)

**S4 Table. Effect of CO$_3^{2-}$ on the degradation of NIF *via* UV/H$_2$O$_2$.**
(DOCX)

**S5 Table. Effect of Cl$^-$ on the degradation of NIF *via* UV/H$_2$O$_2$.**
(DOCX)

**S6 Table. Effect of NO$_3^-$ on the degradation of NIF *via* UV/H$_2$O$_2$.**
(DOCX)

**S7 Table. Molecular weight, molecular formula, structural formula and *m/z* of intermediate products.**
(DOCX)

**S8 Table. Cost analysis of UV/H$_2$O$_2$ treatment.**
(DOCX)

## Author Contributions

**Conceptualization:** Wenping Dong, Chuanxi Yang, Lingli Zhang.

**Data curation:** Wenping Dong, Chuanxi Yang, Lingli Zhang.

**Formal analysis:** Chuanxi Yang, Lingli Zhang, Qiang Su, Xiaofeng Zou.

**Funding acquisition:** Weiliang Wang.

**Investigation:** Qiang Su.

**Methodology:** Qiang Su, Xiaofeng Zou, Kang Xie.

**Software:** Xiaofeng Zou.

**Supervision:** Weiliang Wang.

**Validation:** Wenfeng Xu, Xingang Gao, Kang Xie.

**Visualization:** Wenfeng Xu, Xingang Gao, Kang Xie.

**Writing – original draft:** Wenping Dong, Chuanxi Yang.

**Writing – review & editing:** Chuanxi Yang.

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
