## [Decision Letter · Decision Letter 0]

23 Jul 2021

PONE-D-21-20580

Highly efficient UV/H2O2 technology for the removal of nifedipine antibiotics

PLOS ONE

Dear Dr. Wang,

Thank you for submitting your manuscript to PLOS ONE. After careful consideration, we feel that it has merit but does not fully meet PLOS ONE’s publication criteria as it currently stands. Therefore, we invite you to submit a revised version of the manuscript that addresses the points raised during the review process.

We look forward to receiving your revised manuscript.

Kind regards,

Van-Huy Nguyen, Ph.D.

Academic Editor

PLOS ONE

Journal Requirements:

"This work was supported by the National Natural Science Foundation of China (41672340) and the Research and Demonstration of Special Reagents for Sewage Treatment Plant in Chemical Industry Park (RD28-2019)."

"YES - Specify the role(s) played."

Additional Editor Comments:

There is a concern on your experiment design. No replicates and/or the related appropriate statistics is reported in results and discussion of your manuscript. However, to replicate experiments and to assess the uncertainty of results is at the core of science and indispensable for reliable and high quality results. Repetition of experiments for Figs. 2 to 7 should be done.

Reviewers' comments:

Reviewer's Responses to Questions

**Comments to the Author**

1. Is the manuscript technically sound, and do the data support the conclusions?

Reviewer #1: Yes

Reviewer #2: Yes

Reviewer #3: Partly

Reviewer #4: Partly

2. Has the statistical analysis been performed appropriately and rigorously? 

Reviewer #1: No

Reviewer #2: Yes

Reviewer #3: N/A

Reviewer #4: Yes

3. Have the authors made all data underlying the findings in their manuscript fully available?

Reviewer #1: Yes

Reviewer #2: Yes

Reviewer #3: Yes

Reviewer #4: Yes

4. Is the manuscript presented in an intelligible fashion and written in standard English?

Reviewer #1: No

Reviewer #2: Yes

Reviewer #3: No

Reviewer #4: No

5. Review Comments to the Author

Reviewer #1: The manuscript presented the approach of UV/H2O2 technology towards the nifedipine antibiotics removal. Before considering for publications in this journal, the manuscript needs to consider the following comments

Abstract section contented with lots of data rather highlight the important findings of the present study

In the introduction critical comparison of the different methods for the removal of water pollutant needs to be included and the following references may be useful

https://doi.org/10.1016/j.psep.2021.06.042

https://doi.org/10.1016/j.eti.2021.101581

https://doi.org/10.1016/j.jhazmat.2021.125960

https://doi.org/10.1007/s13399-020-01173-3

https://doi.org/10.1080/10643389.2020.1847949

https://doi.org/10.1016/j.envres.2021.111622

Include the photograph of the experimental setup or present the schematic representation

Better present the effect of process parameters in figure representations

Perform the good number of cycles and present their results

Report the cost analysis of the treatment process

Error bar must be mentioned in all the experimental results presented

Reviewer #2: The article entitled as Highly efficient UV/H2O2 technology for the removal of nifedipine antibiotics: Kinetics, co-existing anion and degradation pathway has scientific value and relevance. It is a timely manuscript. However, there are some gap in this. The following point must be considered before final submission

1) Why homogenous photocatalytic methods is preferred over methods are preferred over semiconductor photocatalyst are concerned. Plz discuss in term of latest work.

https://doi.org/10.1016/j.cej.2019.123496

https://doi.org/10.1016/j.jphotochem.2019.01.015

https://doi.org/10.1016/j.arabjc.2018.12.001

doi:10.5004/dwt.2017.20831

https://doi.org/10.1016/j.jscs.2019.07.003

2. The authors must mention the advantages photocatalysis over other treatment techniques. Plz discuss with reference to pioneer work.

1. https://doi.org/10.1016/j.mtener.2020.100589.

2. https://doi.org/10.1039/D0TA08384D.

3 https://doi.org/10.1016/j.jece.2019.103272. 4

4. https://doi.org/10.1016/j.seppur.2019.115692.

3. The graph must be drawn with good resolution

4 . How the temperature of 20 degree was maintained.

5. The conclusion must be re-written in more conclusive way

Reviewer #3: The manuscript “Highly efficient UV/H2O2 technology for the removal of nifedipine antibiotics”

Using UV/H2O2 technology is popular in removal organic contaminants. The author needs to mention the novel of this study in Introduction section.

Abstract:

- should write fully before using abbreviation, such as: NIF, ROS

- should add the co-existing anions effects

Introduction:

- Please add more removal techniques of NIF, and then find out the advantages and disadvantages of available techniques.

- The author needs to mention the novel of this study in Introduction section.

- Ref 10 & 12 were mentioned in Introduction section. In my opinion, the author reviewed the results of these researches clearly, but they did not release the advantages and disadvantages.

- The objectives of this study should be brief writing. Don’t need to describe the experiment, analysis and reaction pathway in Introduction section.

“Intermediate products analysis and degradation pathways were also obtained based on the Agilant 6520 series liquid chromatogram mass spectrometry (LC-Q-TOF-MS) system. There existed 11 kinds of intermediate products (P345, P329, P329-2, P315, P301, P274, P271, P241, P200, P181 and P158) and 2 degradation pathways (dehydrogenation reaction of NIF → P345 → P274 and dehydration reaction of NIF → P329 → P315)” � should be removed or rewrite.

- Agilant  Agilent

Agilant 6520 series or Agilant 1260 series, please check it.

Materials and Methods:

- In Organics analysis section, please describe clearly the analysis programe

Results and discussion:

- From 3.1 to 3.6, the authors have just listed the results; please give some reasons to help the reader understand these effects.

- Could you give the comparison the effect between co-existing anions?

- Section 3.7 should be brief writing. If possible, move some discussion in this section to 3.1-3.6 sections.

- Author should add text to explain why “the degradation kinetics of NIF via UV/H2O2 system was linear decrease with SO42- concentration increasing.” “the degradation kinetics of NIF via UV/H2O2 system was linear decrease with CO32- concentration increasing” … instead of provide equations E3- E41.

- Please check SO42- in page 24 “The degradation rate of NIF decreases with increasing concentration of SO42- concentration, and the quenching mechanism of ROS via CO32- was shown as following equation 27 (E27) to equation 31 (E31)”

Reviewer #4: In the paper titled “Highly efficient UV/H2O2 technology for the removal of nifedipine antibiotics” authors have done systematic studies on removal of NIF in presence of UV and H2O2. The work is good but it needs some revision

1. The whole article has serious errors with English and the sentence drafting, it is highly urged to fix it.

2. Authors have used EPR studies, more experimental conditions like magnetic field, temperature etc need to be specified. This are very imp. parameters

3. Authors have repeated the data shown in different table in the manuscript as well while explanation, I think this is not comfortable and does not make any sense for the repetition of data. This needs to be removed. Authors should use values from different tables to explain their observations and restricts themselves by writing all the number in manuscript.

4. Authors should perform a study on the effect of NIF removal for different initial concentration of the NIF.

5. Authors have mentioned that pH of the water shows linear decrease with increase in pH value, however as far as I can see the from 4-7 the removal rate is almost constant. How to justify this?

6. PLOS authors have the option to publish the peer review history of their article (what does this mean?). If published, this will include your full peer review and any attached files.

Reviewer #1: **Yes: **Rangabhashiyam S

Reviewer #2: **Yes: **ok

Reviewer #3: No

Reviewer #4: No

---

## [Author Response · Author response to Decision Letter 0]

9 Sep 2021

Response to Editor's comments:

Thank you very much for your consideration of our manuscript entitled “Highly efficient UV/H2O2 technology for the removal of nifedipine antibiotics: Kinetics, co-existing anions and degradation pathways” (PONE-D-21-20580) to Plos One.

We have read the editor’s comments and reviewers’ questions carefully, and made an attempt to respond as clearly as possible. In reference to the above recommendations, the responses are shown in “Response to 1st reviewer”, “Response to 2nd reviewer”, “Response to 3rd reviewer” and “Response to 4th reviewer”.

Response to 1st reviewer:

Question 1: In the introduction critical comparison of the different methods for the removal of water pollutant needs to be included and the following references may be useful.

https://doi.org/10.1016/j.psep.2021.06.042

https://doi.org/10.1016/j.eti.2021.101581

https://doi.org/10.1016/j.jhazmat.2021.125960

https://doi.org/10.1007/s13399-020-01173-3

https://doi.org/10.1080/10643389.2020.1847949

https://doi.org/10.1016/j.envres.2021.111622

Answer.

Thanks a lot. In [Introduction], the critical comparison of the different methods for the removal of water pollutant was added. And the references ([13], [14], [15], [16], [17] and [18]) were cited.

Adsorption and advanced oxidation processes (AOPs, such as photocatalysis, Fenton, Fenton-like, photo-Fenton and catalytic ozonation) are the most promising wastewater treatment technologies for the removal of antibiotics from water environments and reduction of the resulting environmental risks because they are fast, efficient, low cost and convenient[13-16]. Many adsorbents have been employed for the eradication of antibiotics[17]. However, there are some drawbacks , such as incomplete removal, high energy requirements and the generation of toxic sludge and other waste products that entail further disposal[18].

[13] Hasija V, Raizada P, Singh P, Verma N, Khan AAP, Singh A, Selvasembian R, Kim SY, Hussain CM, Nguyen VH, Le QV. Progress on the photocatalytic reduction of hexavalent Cr (VI) using engineered graphitic carbon nitride. Process Saf Environ. 2021; 152: 663-678.

[14] Soni V, Raizada P, Singh P, Cuong HN, Rangabhashiyam S, Saini A, Saini RV, Le QV, Nadda AK, Le TT, Nguyen VH. Sustainable and green trends in using plant extracts for the synthesis of biogenic metal nanoparticles toward environmental and pharmaceutical advances: A review. Environ Res. 2021; 202: 111622.

[15] Gunarathne V, Rajapaksha AU, Vithanage M, Alessi DS, Selvasembian R, Naushad M, You SM, Oleszczuk P, Ok YS. Hydrometallurgical processes for heavy metals recovery from industrial sludges. Crit Rev Env Sci Tec. 2020; 10.1080/10643389.2020.1847949.

[16] Hariharan A, Harini1 V, Sandhya S, Rangabhashiyam S. Waste Musa acuminata residue as a potential biosorbent for the removal of hexavalent chromium from synthetic wastewater. Biomass Convers Bior. 2020; 10.1007/s13399-020-01173-3.

[17] Rangabhashiyam S, Vijayaraghavan K, Jawad AH, Singh P, Singh P. Sustainable approach of batch and continuous biosorptive systems for praseodymium and thulium ions removal in mono and binary aqueous solutions. Environ Technol Inno. 2021; 23: 101581.

[18] Selvasembian R, Gwenzi W, Chaukura N , Mthembu S. Recent advances in the polyurethane-based adsorbents for the decontamination of hazardous wastewater pollutants. J Hazard Mater. 2021; 417: 125960.

Question 2: Include the photograph of the experimental setup or present the schematic representation.

Answer.

The photograph of the experimental setup was added as Fig. 2 in [Experimental setup].

Fig. 2 Experimental setup of UV/H2O2.

Question 3: Better present the effect of process parameters in figure representations.

Answer.

The effect of H2O2 dose, initial pH and co-existing anions (SO42-, CO32-, Cl- and NO3-) is presented as Fig.3, Fig.4, Fig.5, Fig.6, Fig.7 and Fig.8 in [Results and discussion].

Fig. 3 Effect of H2O2 dose on removal rate (a), kinetics constant (b) and linear fitting between kinetics constant and H2O2 dose on the degradation of NIF via UV/H2O2. The error bars represent the standard deviation (n = 3).

Fig. 4 Effect of initial pH on the removal rate (a), kinetics constant (b) and linear fitting between kinetics constant and initial pH on the degradation of NIF via UV/H2O2. The error bars represent the standard deviation (n = 3).

Fig. 5 Effect of SO42- on the removal rate (a), kinetics constant (b) and linear fitting between kinetics constant and concentration of SO42- on the degradation of NIF via UV/H2O2. The error bars represent the standard deviation (n = 3).

Fig. 6 Effect of CO32- on the removal rate (a), kinetics constant (b) and linear fitting between kinetics constant and concentration of CO32- on the degradation of NIF via UV/H2O2. The error bars represent the standard deviation (n = 3).

Fig. 7 Effect of CO32- on the removal rate (a), kinetics constant (b) and linear fitting between kinetics constant and concentration of CO32- on the degradation of NIF via UV/H2O2. The error bars represent the standard deviation (n = 3).

Fig. 8 Effect of NO3- on the removal rate (a), kinetics constant (b) and linear fitting between kinetics constant and concentration of NO3- on the degradation of NIF via UV/H2O2. The error bars represent the standard deviation (n = 3).

Question 4: Perform the good number of cycles and present their results.

Answer.

The UV/H2O2 system was not evaluated the recycleability due to the unrecoverable ability of H2O2. And we retrievaled the references of UV/H2O2 technology, the cyclic test of degradation of nifedipine via the UV/H2O2 method was not designed successfully.

Question 5: Report the cost analysis of the treatment process.

Answer.

The cost analysis of UV/H2O2 system was added in [Environmental significance].

In addition, the cost of the UV/H2O2 system was approximately $0.447 for 1 m3 wastewater (Table S8), which was lower than that of related systems (ranging from $0.53 to $0.85 for 1 m3 wastewater)[47].

Table S8. Cost analysis of UV/H2O2 treatment.

Materials Prices Price for treatment of 1 m3 wastewater

25 W UV lamp $17.8 for 9000 h $0.165

H2O2 $4.2 for 1 kg $0.074

Power $0.1 for 1 kW•h $0.208

Total prices $22.1 $0.447

Reference 47 / [$0.53, $0.85]

Question 6: Error bar must be mentioned in all the experimental results presented.

Answer.

The errer bar was added as Fig.3, Fig.4, Fig.5, Fig.6, Fig.7 and Fig.8 in [Results and discussion].

See the answers of Question 3.

Response to 2nd reviewer:

Question 7: Why homogenous photocatalytic methods is preferred over methods are preferred over semiconductor photocatalyst are concerned. Please discuss in term of latest work.

https://doi.org/10.1016/j.cej.2019.123496

https://doi.org/10.1016/j.jphotochem.2019.01.015

https://doi.org/10.1016/j.arabjc.2018.12.001

doi:10.5004/dwt.2017.20831

https://doi.org/10.1016/j.jscs.2019.07.003

Answer.

The comparison between homogenous photocatalyst and semiconductor photocatalyst was added in [Introduction]. And the references ([23], [24], [25], [26] and [27]) were cited.

However, the limitations of a wide bandgap, the rapid recombination rate of photogenerated electron-hole pairs, low solar light energy utilization efficiency, photocorrosion, and poor recyclability reduce the photocatalytic efficiency[23-25]. It is imperative to develop a novel Z-scheme system or heterojunction photocatalyst with broad photocatalytic applications[26-27].

[23] Kumar A, Raizada P, Singh P, Saini RV, Saini AK, Hosseini-Bandegharaei A. Perspective and status of polymeric graphitic carbon nitride based Z-scheme photocatalytic systems for sustainable photocatalytic water purifcation. Chem Eng J. 2020; 391: 123496.

[24] Singh P, Shandilya P, Raizada P, Sudhaik A, Rahmani-Sani A, Hosseini-Bandegharaei A. Review on various strategies for enhancing photocatalytic activity of graphene based nanocomposites for water purification. Arab J Chem. 2020; 13: 3498-3520.

[25] Raizada P, Kumari J, Shandilya P, Singh P. Kinetics of photocatalytic mineralization of oxytetracycline and ampicillin using activated carbon supported ZnO/ZnWO4 nanocomposite in simulated wastewater. Desalin Water Treat. 2017; 79: 204-213.

[26] Raizada P, Sudhaik A, Singh P, Shandilya P, Gupta VK, Hosseini-Bandegharaei A, Agrawal S. Ag3PO4 modified phosphorus and sulphur co-doped graphitic carbon nitride as a direct Z-scheme photocatalyst for 2, 4-dimethyl phenol degradation. J Photoch Photobio A. 2019; 374: 22-35.

[27] Sonu, Dutta V, Sharma S, Raizada P, Hosseini-Bandegharaei A, Gupta VK, Singh P. Review on augmentation in photocatalytic activity of CoFe2O4 via heterojunction formation for photocatalysis of organic pollutants in water. J Saudi Chem Soc. 2019; 23: 1119-1136.

Question 8: The authors must mention the advantages photocatalysis over other treatment techniques. Please discuss with reference to pioneer work.

https://doi.org/10.1016/j.mtener.2020.100589.

https://doi.org/10.1039/D0TA08384D.

https://doi.org/10.1016/j.jece.2019.103272.4

https://doi.org/10.1016/j.seppur.2019.115692

Answer.

The advantages of photocatalysis were added in [Introduction]. And the references ([19], [20], [21] and [22]) were cited.

Solar light-driven photocatalysis involves the photoinduced generation of holes (h+) in the valence band (VB) and electrons (e-) in the conduction band (CB) via light absorption by a semiconductor (TiO2, ZnO and CdS). Sequential interfacial charge transfers release various reactive oxygen species (ROS), such as superoxide, peroxide, and hydroxyl radicals, which participate in the degradation of organic and inorganic pollutants[19-22].

[19] Kumar A, Raizada P, Hosseini-Bandegharaei A, Thakur VK, Nguyen VH, Singh P. C-, N-vacancies defect engineering polymeric carbon nitride meeting photocatalysis: Viewpoints and challenges. J Mater Chem A. 2020; 9: 111-153.

[20] Raizada P, Sudhaik A, Singh P, Hosseini-Bandegharaei A, Thakur P. Converting type II AgBr/VO into ternary Z scheme photocatalyst via coupling with phosphorus doped g-C3N4 for enhanced photocatalytic activity. Sep Purif Technol. 2019; 227: 115692.

[21] Hasija V, Sudhaik A, Raizada P, Hosseini-Bandegharaei A, Singh P. Carbon quantum dots supported AgI/ZnO/phosphorus doped graphitic carbon nitride as Z-scheme photocatalyst for efficient photodegradation of 2, 4-dinitrophenol. J Environ Chem Eng. 2019; 7: 103272.

[22] Patial S, Raizada P, Hasija V, Singh P, Thakur VK, Nguyen VH. Recent advances in photocatalytic multivariate metal organic frameworks-based nanostructures toward renewable energy and the removal of environmental pollutants. Mater Today Energy. 2021; 19: 100589.

Question 9: The graph must be drawn with good resolution

Answer.

All the figures were drawn with good resolution of 300 dpi.

Question 10: How the temperature of 20 degree was maintained.

Answer.

The temperature of 20 degree was maintained using thermostat water bath oscillator as Fig. 2 in [Experimental setup].

Fig. 2 Experimental setup of UV/H2O2.

Question 11: The conclusion must be re-written in more conclusive way.

Answer.

The conclusion was re-written as follows:

The degradation rate and degradation kinetics of NIF first increased and then remained constant as the H2O2 dose increased, and the quasi-percolation threshold was an H2O2 dose of 0.378 mmol/L. The effect of the initial pH, divalent anions (SO42- and CO32-) and monovalent anions (Cl- and NO3-) decreased linearly with increasing initial pH and co-existing anions (the R2 values of the initial pH, SO42-, CO32- Cl- and NO3- were 0.6884, 0.9939, 0.8589, 0.5013 and 0.4514, respectively). •OH was the primary key ROS, and •O2- was the secondary key ROS. There were 11 intermediate products (P345, P329, P329–2, P315, P301, P274, P271, P241, P200, P181 and P158) and 2 degradation pathways (dehydrogenation reaction of NIF → P345 → P274 and dehydration reaction of NIF → P329 → P315).

Response to 3rd reviewer:

Question 12: Abstract:

- should write fully before using abbreviation, such as: NIF, ROS

Answer.

The full name of NIF and ROS was added in [Abstract].

This study investigates the degradation of nifedipine (NIF) by using a novel and highly efficient ultraviolet light combined with hydrogen peroxide (UV/H2O2).

The •OH was the primary key reactive oxygen species (ROS) and •O2- was the secondary key ROS.

Question 13: Abstract:

- should add the co-existing anions effects

Answer.

The co-existing anions effects were added in [Abstract].

An increase in the initial pH and divalent anions (SO42- and CO32-) resulted in a linear decrease of NIF (the R2 of the initial pH, SO42- and CO32- was 0.6884, 0.9939 and 0.8589, respectively). The effect of monovalent anions was complex; Cl- and NO3- had opposite effects: low Cl- or high NO3- promoted degradation, and high Cl- or low NO3- inhibited the degradation of NIF.

Question 14: Introduction:

- Please add more removal techniques of NIF, and then find out the advantages and disadvantages of available techniques.

Answer.

The removal techniques of NIF were added in [Introduction].

However, limited research has focused on the removal of NIF from the water environment via AOPs. Therefore, it is important to study the removal of NIF via AOPs for the treatment of medical wastewater.

Mojtaba Shamsipur et al. used a multivariate curve resolution method based on the combination of the Kubista approach and an iterative target transformation method by Gemperline to study the kinetics of NIF decomposition upon exposure to a 40 W lamp[29]. The results indicated that the photodecomposition kinetics of NIF are zero-order at the beginning of the reaction. However, when the reaction was more than 50% complete, the kinetics of the reaction changed to a first-order mechanism. The photo-degradation kinetics constants for the zero-order and first-order regions were (4.96±0.13)*10-9 M-1 s-1 and (6.22±0.10)*10-5 s-1, respectively. This was the first study on the degradation of NIF, but the low degradation rate (65%) and kinetics limited the application of NIF removal via a photo-degradation system.

Question 15: Introduction:

- The author needs to mention the novel of this study in Introduction section.

Answer.

The novel of this study was mentioned in [Introduction].

A novel method of UV light combined with hydrogen peroxide (UV/H2O2) is highly efficient, fast, and has a strong oxidizing ability; these advantages are attributed to the synergistic ability of UV light and H2O2 to generate ROS[30]. In our previous study, the degradation of norfloxacin by using UV/H2O2 was investigated[31]. The degradation rate and apparent first-order kinetics constant of norfloxacin via UV/H2O2 were 98.8% and 0.22248 min-1, respectively, and the norfloxacin concentration = 20 mg/L, the H2O2 dose =1.2 mmol/L, the pH=7, T= 20 °C and the reaction time = 20 min. The kinetics were low, and the formation mechanism of ROS was controversial, but it provided a novel research direction for the degradation of NIF via a UV/H2O2 system. Therefore, it should be noted that the degree of research to date on the degradation of NIF via UV/H2O2 oxidation processes is insufficient to thoroughly understand the fundamentals of •OH generation, intermediate products and degradation pathways, which are important processes that must be considered in the design of wastewater treatment technology[32]. Furthermore, the effect of co-existing anions in the UV/H2O2 system may significantly affect the removal of NIF by competitively quenching with ROS[33]. Thus, it is still challenging to design a UV/H2O2 wastewater treatment technology with high efficiency.

Question 16: Introduction:

- Ref 10 & 12 were mentioned in Introduction section. In my opinion, the author reviewed the results of these researches clearly, but they did not release the advantages and disadvantages.

Answer.

The advantages and disadvantages of reference 29 (Ref 10 of original manuscript) and 31(Ref 12 of original manuscript) were added in [Introduction].

Mojtaba Shamsipur et al. used a multivariate curve resolution method based on the combination of the Kubista approach and an iterative target transformation method by Gemperline to study the kinetics of NIF decomposition upon exposure to a 40 W lamp[29]. The results indicated that the photodecomposition kinetics of NIF are zero-order at the beginning of the reaction. However, when the reaction was more than 50% complete, the kinetics of the reaction changed to a first-order mechanism. The photo-degradation kinetics constants for the zero-order and first-order regions were (4.96±0.13)*10-9 M-1 s-1 and (6.22±0.10)*10-5 s-1, respectively. This was the first study on the degradation of NIF, but the low degradation rate (65%) and kinetics limited the application of NIF removal via a photo-degradation system.

In our previous study, the degradation of norfloxacin by using UV/H2O2 was investigated[31]. The degradation rate and apparent first-order kinetics constant of norfloxacin via UV/H2O2 were 98.8% and 0.22248 min-1, respectively, and the norfloxacin concentration = 20 mg/L, the H2O2 dose =1.2 mmol/L, the pH=7, T= 20 °C and the reaction time = 20 min. The kinetics were low, and the formation mechanism of ROS was controversial, but it provided a novel research direction for the degradation of NIF via a UV/H2O2 system.

Question 17 Introduction:

- The objectives of this study should be brief writing. Don’t need to describe the experiment, analysis and reaction pathway in Introduction section.

Answer.

The objectives of this study were re-written in [Introduction].

The aims of this study were to demonstrate the application of NIF degradation and to evaluate the performance and mechanism of UV/H2O2 AOPs. The specific objectives were (1) to assess the effect of the H2O2 dose, initial pH, and co-existing anions (SO42-, CO32-, Cl- and NO3-) on the degradation of UV/H2O2, (2) to predict the key ROS of the UV/H2O2 method and (3) to propose the degradation pathway of NIF.

Question 18 Introduction:

- “Intermediate products analysis and degradation pathways were also obtained based on the Agilant 6520 series liquid chromatogram mass spectrometry (LC-Q-TOF-MS) system. There existed 11 kinds of intermediate products (P345, P329, P329-2, P315, P301, P274, P271, P241, P200, P181 and P158) and 2 degradation pathways (dehydrogenation reaction of NIF → P345 → P274 and dehydration reaction of NIF → P329 → P315)” should be removed or rewrite.

Answer.

The objectives of this study were re-written in [Introduction] and this sentences were removed.

Question 19 Introduction:

- Agilant  Agilent

- Agilant 6520 series or Agilant 1260 series, please check it.

Answer.

“Agilant” was changed to “Agilent” in the manuscript.

In this study, the Agilent 1260 series liquid chromatogram mass spectrometry (LC-Q-TOF-MS) system (Agilent, USA) was used.

Question 20 Materials and Methods:

- In Organics analysis section, please describe clearly the analysis programe.

Answer.

NIF and its intermediate products in the UV/H2O2 degradation reaction solutions were analyzed by an Agilent 1260 series liquid chromatogram mass spectrometry (LC-Q-TOF-MS) system (Agilent, USA) with a C18 column (100 mm × 2.1 mm, 3.5 mm). The wavelength was 237 nm according to ultraviolet and visible spectrophotometry (Fig. S1). The mobile phase was methyl alcohol and deionized water at 63:35 (v/v). The drying gas of N2 was 8.0 mL/min, and the testing time was 30 min.

Question 21 Results and discussion:

- From 3.1 to 3.6, the authors have just listed the results; please give some reasons to help the reader understand these effects.

Answer.

The reasons for degradation of NIF was added in [Results and discussion].

For [3.1 Effect of H2O2 dose] as following:

As shown in Fig. 3c, the effect of the H2O2 dose on the degradation of NIF via UV/H2O2 system had a dual nature. The degradation kinetics constant noticeably increased as the H2O2 dose increased and then remained constant at 1.5±0.1 min-1. When the H2O2 dose was < 0.52 mmol/L, the slope was 3.013 (min-1)/(mmol/L), but it decreased to 0.266 (min-1)/(mmol/L) when the H2O2 dose was > 0.52 mmol/L; hence, the quasi-percolation threshold (QPT) of the H2O2 dose was 0.378 mmol/L[34]. This trend was based on the generation and quenching of •OH described by Equations 3 (E3) to 6 (E6)[35]:

(a) H2O2 dose < QPT:

H2O2 → 2 •OH (E3)

(b) H2O2 dose > QPT:

H2O2 + •OH → •HO2 + H2O (E4)

•HO2 + •OH → H2O + O2 (E5)

2 •OH → H2O2 (E6)

Fig. 3 Effect of H2O2 dose on removal rate (a), kinetics constant (b) and linear fitting between kinetics constant and H2O2 dose on the degradation of NIF via UV/H2O2. The error bars represent the standard deviation (n = 3).

For [3.2 Effect of initial pH] as following:

As shown in Fig. 4c, the degradation kinetics of NIF via UV/H2O2 system exhibited a poor linear decrease as the pH increased (y = -0.1155x + 1.95533, R2 = 0.6884). The results indicated that acidic solution (pH = 4) was more favorable for degrading NIF than basic solution (pH = 10) under UV/H2O2. The possible reasons were in accord with the redox potential, generation rate of ROS and reaction rate between ROS and NIF. The inhibiting effect of the basic solution was due to the quenching reaction between OH- and •OH, which is shown in Equations 7 (E7) to 11 (E11)[36]:

H2O2 → HO2- + H+ (E7)

H+ + OH- → H2O (E8)

H2O2 + HO2- → H2O + OH- + O2 (E9)

•OH + HO2- → OH- + •HO2 (E10)

•OH + HO2- → H2O + •O2- (E11)

Fig. 4 Effect of initial pH on the removal rate (a), kinetics constant (b) and linear fitting between kinetics constant and initial pH on the degradation of NIF via UV/H2O2. The error bars represent the standard deviation (n = 3).

For [3.3 Effect of SO42-] as following:

As shown in Fig. 5c, the degradation kinetics of NIF via UV/H2O2 system exhibited a good linear decrease as the SO42- concentration increased (y = -0/0088x + 1.437, R2 = 0.9939). The inhibition effect was 5.08% (SO42- concentration = 5 mg/L) < 14.45% (SO42- concentration = 20 mg/L) < 31.20% (SO42- concentration = 50 mg/L), which was in keeping with the trend of the kinetics constants: 1.00154 min-1 (SO42- concentration = 50 mg/L) < 1.24540 min-1 (SO42- concentration = 20 mg/L) < 1.38175 min-1 (SO42- concentration = 5 mg/L). The degradation rate of NIF decreased with increasing SO42- concentration, and the quenching mechanism of ROS via SO42- is shown in Equations 12 (E12) to 17 (E17)[37]:

H+ + SO42- → HSO4- (E12)

HSO4- + •OH → SO4•- + H2O (E13)

SO4•- + H2O → H+ + SO42- + •OH (E14)

SO4•- + H2O2 → H+ + SO42- + H2O• (E15)

SO4•- + H2O• → H+ + SO42- + O2 (E16)

SO4•- + e- → SO42- (E17)

Fig. 5 Effect of SO42- on the removal rate (a), kinetics constant (b) and linear fitting between kinetics constant and concentration of SO42- on the degradation of NIF via UV/H2O2. The error bars represent the standard deviation (n = 3).

For [3.4 Effect of CO32-] as following:

As shown in Fig. 6c, the degradation kinetics of NIF via the UV/H2O2 system exhibited a good linear decrease as the CO32- concentration increased (y = -0.0072x + 1.3771, R2 = 0.8589). The inhibition trend was 11.00% (CO32- concentration = 5 mg/L) < 19.69% (CO32- concentration = 20 mg/L) < 28.25% (CO32- concentration = 50 mg/L), which was in keeping with that of kinetics constant: 1.04447 min-1 (CO32- concentration = 50 mg/L) < 1.16907 min-1 (CO32- concentration = 20 mg/L) < 1.29550 min-1 (CO32- concentration = 5 mg/L). The degradation rate of NIF decreased with increasing CO32- concentration, and the quenching mechanism of the ROS via CO32- is shown in Equations 18 (E18) to 22 (E22)[38]:

CO32- + H+ → HCO3- (E18)

HCO3- → CO32- + H+ (E19)

CO32- + •OH → CO3•- + OH- (E20)

HCO3- + •OH → CO3•- + H2O (E21)

CO3•- + H2O2 → HO2• + HCO3- (E22)

Fig. 6 Effect of CO32- on the removal rate (a), kinetics constant (b) and linear fitting between kinetics constant and concentration of CO32- on the degradation of NIF via UV/H2O2. The error bars represent the standard deviation (n = 3).

For [3.5 Effect of Cl-] as following:

As shown in Fig. 7c, although the degradation kinetics of NIF via UV/H2O2 system decreased with increasing Cl- concentration, the effect of Cl- on the degradation of NIF had a dual nature: low Cl- concentrations promoted the degradation of NIF, while high Cl- concentrations inhibited the degradation of NIF. The degradation kinetics of NIF via UV/H2O2 system exhibited a poor linear decrease as the Cl- concentration increased (y = -0.012x + 1.7897, R2 = 0.5013). The trend of inhibition was -36.25% (Cl- concentration = 5 mg/L) < -18.61% (Cl- concentration = 20 mg/L) < 24.71% (Cl- concentration = 50 mg/L), which was in keeping with the trend of kinetics constant: 1.09588 min-1 (Cl- concentration = 50 mg/L) < 1.72666 min-1 (Cl- concentration = 20 mg/L) < 1.98350 min-1 (Cl- concentration = 5 mg/L). The reaction mechanism between Cl- and •OH is shown in Equations 23 (E23) to 27 (E27)[39]:

Cl- + •OH → Cl• + OH- (E23)

Cl• + Cl- → •Cl2- (E24)

•Cl2- + H2O2 → H+ + 2 Cl- + H2O• (E25)

•OH + Cl- → •HOCl- (E26)

•HOCl- + H+ → H2O + Cl- (E27)

Fig. 7 Effect of CO32- on the removal rate (a), kinetics constant (b) and linear fitting between kinetics constant and concentration of CO32- on the degradation of NIF via UV/H2O2. The error bars represent the standard deviation (n = 3).

For [3.6 Effect of NO3-] as following:

As shown in Fig. 8c, NO3- had the opposite effect on the degradation of NIF via the UV/H2O2 system compared to Cl-. The effect of NO3- on the degradation of NIF had a dual nature: low NO3- concentrations inhibited the degradation of NIF, but high NO3- concentrations promoted the degradation of NIF. The degradation kinetics of NIF via the UV/H2O2 system showed a poor linear decrease with increasing NO3- concentration (y = -0.005377x + 1.31881, R2 = 0.4514). The inhibition effect was -6.68% (NO3- concentration = 50 mg/L) < 10.83% (NO3- concentration = 20 mg/L) < 29.10% (NO3- concentration = 5 mg/L), which was in keeping with the trend of the kinetics constant: 1.03215 min-1 (NO3- concentration = 5 mg/L) < 1.29801 min-1 (NO3- concentration = 20 mg/L) < 1.55295 min-1 (NO3- concentration = 50 mg/L). The mechanism of the reaction between NO3- and •OH is shown in Equations 28 (E28) to 32 (E32)[40]:

NO3- + hv → NO2- + O (E28)

NO3- + hv → NO2• + O•- (E29)

2 NO2• + H2O → NO2- + NO - + 2 H+ (E30)

O + H2O → 2 •OH (E31)

O•- + H2O → •OH + OH- (E32)

Fig. 8 Effect of NO3- on the removal rate (a), kinetics constant (b) and linear fitting between kinetics constant and concentration of NO3- on the degradation of NIF via UV/H2O2. The error bars represent the standard deviation (n = 3).

Question 22 Results and discussion:

- Could you give the comparison the effect between co-existing anions?

Answer.

The comparison of co-existing anions for degradation of NIF was added in [Results and discussion].

In summary, the effect of co-existing anions was as follows: The divalent anions (SO42- and CO32-) caused a good linear decrease with increasing initial pH and concentration of divalent anions (R2 of SO42- and CO32- was 0.9939 and 0.8589, respectively). The monovalent anions had a complex effect; Cl- and NO3- had opposite effects on NIF degradation: low Cl- and high NO3- promoted degradation, while high Cl- and low NO3- inhibited degradation. The degradation kinetics of NIF via the UV/H2O2 system showed a poor linear decrease with increasing Cl-/NO3- concentration (R2 of Cl- and NO3- was 0.5013 and 0.4514, respectively).

Question 23 Results and discussion:

- Section 3.7 should be brief writing. If possible, move some discussion in this section to 3.1-3.6 sections.

Answer.

The [3.7 Oxidation mechanism and degradation pathway of NIF via UV/H2O2] was re-written. The related discussion in this section was moved to 3.1-3.6 sections as the answers of Question 21.

In recent advances in UV/H2O2 systems, the degradation of organic pollutants has been due to the generation of ROS, especially •OH and •O2-. The oxidation mechanism of NIF degradation via the UV/H2O2 system was measured by ESR measurements, and the ESR spectra are shown in Fig. 9(a-b). The significant •OH signal (Fig. 9a) showed four peaks at 321.8 mT (PA), 323.3 mT (PB), 324.8 mT (PC) and 326.3 mT (PD). The interspaces of PA-PB, PB-PC and PC-PD were constant of 1.5 mT, and the intensity ratio of PA, PB, PC and PD was 1:2:2:1[41]. The significant •O2- signal (Fig. 9b) showed four peaks at 322.1 mT (P’A), 323.2 mT (P’B), 324.4 mT (P’C) and 325.8 mT (PD). The interspaces of P’A-P’B, P’B-P’C and P’C-P’D were constant within the range of 1.0 mT from 1.5 mT, and the intensity ratio of P’A, P’B, P’C and P’D was 1:1:1:1[42]. However, the intensity of the •OH signal was stronger than that of the •O2- signal, which means that •OH was the primary key ROS and •O2- was the secondary key ROS. The oxidation mechanism of NIF degradation via the UV/H2O2 system is shown in Equations 33 (E33) to 41 (E41)[43]. The generation of •OH mainly comes from the direct decomposition of H2O2 (E33), the generation of •O2- mainly comes from the indirect reaction between oxygen gas (dissolved oxygen and H2O2 decomposition) and electrons (E35 and E38), and NIF is degraded via the ROS (E41)[44].

H2O2 + hv → 2 •OH (E33)

H2O2 + •OH → •HO2 + H2O (E34)

•HO2 → •O2- + H+ (E35)

2 •HO2 → H2O2 + O2 (E36)

•HO2 + •OH → H2O + O2 (E37)

O2 + e- → •O2- (E38)

2 •O2- → 1O2 + 3O2 + 2 e- (E39)

2 •OH → H2O2 (E40)

NIF + ROS → Intermediate Products → CO2 + H2O (E41)

Fig. 9 ESR measurements of •OH (a) and •O2- (b).

The identification of intermediate NIF products via UV/H2O2 was performed using an Agilent 1260 series liquid chromatogram mass spectrometry (LC-Q-TOF-MS) system. The reaction conditions were: NIF concentration = 5 mg/L, pH = 7, H2O2 dose = 0.52 mmol/L, T = 20 °C and reaction time = 30 min. Preliminary analyses were conducted to evaluate the intermediate products of NIF produced by the UV/H2O2 method, and four kinds of NIF degradation intermediate products were observed in the mass spectrum based on the mass-to-charge (m/z) ratio. Four intermediate products were present: P345 (m/z = 345), P329 (m/z = 329), P315 (m/z = 315) and P274 (m/z = 274). The structures of P345, P329, P315 and P274 are shown in Fig. S3a, Fig. S3b, Fig. S3c and Fig. S3d, respectively.

Fig. 10 Chromatography of the intermediate products of NIF via UV/H2O2.

As shown in Fig. 10(a-b), the peak areas of all the observed products first increased and then decreased within 30 min. The peak areas of P345 and P274 at different reaction times are shown in Fig. 10a, and the results indicated that the maximum peak area of P345 appeared at 5 min, the maximum peak area of P274 appeared at 20 min, and the peak intensity of P345 was stronger than that of P274. The first degradation pathway of NIF via the UV/H2O2 method was proposed as follows: protonated NIF (C17H19N2O6+, m/z = 347) first lost H2 to generate P345 (C17H17N2O6+, m/z = 345) via dehydrogenation reaction, and then P345 lost C3H5NO to generate P274 (C14H12NO5+, m/z = 274), as shown in Fig. 11a[45-46]. The peak areas of P329 and P315 at different reaction times are shown in Fig. 10b, and the results indicated that the maximum peak area of P329 appeared at 0.5 min, the maximum peak area of P315 appeared at 7 min, and the peak intensity of P329 was stronger than that of P315. The second degradation pathway of NIF via the UV/H2O2 method was proposed as follows: protonated NIF (C17H19N2O6+, m/z = 347) first lost H2O to generate P329 (C17H17N2O5+, m/z = 329) via dehydration reaction, and then P329 lost CH2 to generate P315 (C16H15N2O5+, m/z = 274), as shown in Fig. 11b[45-46].

Fig. 11 Degradation pathway of NIF via UV/H2O2 system.

P315 was degraded with the attack of ROS. The intermediate products of P271 (m/z = 271), P241 (m/z = 241) and P181 (m/z = 181) are shown in Table S7, and its possible degradation pathway is shown in Fig. S4a. P315 first lost CO2 to generate P271 (C15H15N2O3+, m/z = 271), P271 lost CH2O to generate P241 (C14H13N2O2+, m/z = 241), and finally, P241 lost C2H4O2 to generate P181 (C12H9N2+, m/z = 181)[46]. P329 was present as an isomeride that was named P329-2 (m/z = 329). The intermediate products, P329-2, P301 (m/z = 301), P200 (m/z = 200) and P158 (m/z = 158), are shown in Table S7, and their possible degradation pathways are shown in Fig. S4b. P329-2 first lost CO to generate P301 (C16H17N2O4+, m/z = 301), P301 lost C4H7NO2 to generate P200 (C12H10NO2+, m/z = 200), and finally, P200 lost C2H2O to generate P158 (C10H8NO+, m/z = 158)[46].

Question 24 Results and discussion:

- Author should add text to explain why “the degradation kinetics of NIF via UV/H2O2 system was linear decrease with SO42- concentration increasing.” “the degradation kinetics of NIF via UV/H2O2 system was linear decrease with CO32- concentration increasing” … instead of provide equations E3- E41.

Answer.

The degradation rate of NIF decreased with increasing divalent anions (SO42- and CO32-) was due to the quenching effect. See the answers of Question 21.

Question 25 Results and discussion:

- Author should add text to explain why “the degradation kinetics of NIF via UV/H2O2 system was linear decrease with SO42- concentration increasing.” “the degradation kinetics of NIF via UV/H2O2 system was linear decrease with CO32- concentration increasing” … instead of provide equations E3- E41.

Answer.

See the answers of Question 21.

Response to 4th reviewer

Question 26 The whole article has serious errors with English and the sentence drafting, it is highly urged to fix it.

Answer.

The language of manuscript (grammar, punctuation, spelling, and overall style) was corrected by the Nature Research Editing Service (website: https://secure.authorservices.springernature.com/). The “Editing Certificate” with a verification code [0C36-F874-B639-2E32-1B7F] was used to verify by secure.authorservices.springernature.com/certificate/verify. The detailed changes of this manuscript were in “Manuscript”, “Revised Manuscript with Track Changes” and “Supporting Information”.

Question 27 Authors have used EPR studies, more experimental conditions like magnetic field, temperature etc need to be specified. This are very important parameters.

Answer.

The ESR measurements was in [2.5 Electron spin resonance (ESR) measurements] and [3.7 Oxidation mechanism and degradation pathway of NIF via UV/H2O2].

For [2.5 Electron spin resonance (ESR) measurements]

ESR measurements were performed with a JES-FA200 electron spin resonance spectrometer and used to measure the hydroxide radical (•OH) and superoxide radical (•O2-) during the degradation of NIF under UV/H2O2 using 5,5-dimethyl-1-pyrroline N-oxide (DMPO) as the spin trapping reagent.

For [3.7 Oxidation mechanism and degradation pathway of NIF via UV/H2O2]

3.7 Oxidation mechanism and degradation pathway of NIF via UV/H2O2

In recent advances in UV/H2O2 systems, the degradation of organic pollutants has been due to the generation of ROS, especially •OH and •O2-. The oxidation mechanism of NIF degradation via the UV/H2O2 system was measured by ESR measurements, and the ESR spectra are shown in Fig. 9(a-b). The significant •OH signal (Fig. 9a) showed four peaks at 321.8 mT (PA), 323.3 mT (PB), 324.8 mT (PC) and 326.3 mT (PD). The interspaces of PA-PB, PB-PC and PC-PD were constant of 1.5 mT, and the intensity ratio of PA, PB, PC and PD was 1:2:2:1[41]. The significant •O2- signal (Fig. 9b) showed four peaks at 322.1 mT (P’A), 323.2 mT (P’B), 324.4 mT (P’C) and 325.8 mT (PD). The interspaces of P’A-P’B, P’B-P’C and P’C-P’D were constant within the range of 1.0 mT from 1.5 mT, and the intensity ratio of P’A, P’B, P’C and P’D was 1:1:1:1[42]. However, the intensity of the •OH signal was stronger than that of the •O2- signal, which means that •OH was the primary key ROS and •O2- was the secondary key ROS. The oxidation mechanism of NIF degradation via the UV/H2O2 system is shown in Equations 33 (E33) to 41 (E41)[43]. The generation of •OH mainly comes from the direct decomposition of H2O2 (E33), the generation of •O2- mainly comes from the indirect reaction between oxygen gas (dissolved oxygen and H2O2 decomposition) and electrons (E35 and E38), and NIF is degraded via the ROS (E41)[44].

H2O2 + hv → 2 •OH (E33)

H2O2 + •OH → •HO2 + H2O (E34)

•HO2 → •O2- + H+ (E35)

2 •HO2 → H2O2 + O2 (E36)

•HO2 + •OH → H2O + O2 (E37)

O2 + e- → •O2- (E38)

2 •O2- → 1O2 + 3O2 + 2 e- (E39)

2 •OH → H2O2 (E40)

NIF + ROS → Intermediate Products → CO2 + H2O (E41)

Fig. 9 ESR measurements of •OH (a) and •O2- (b).

Question 28 Authors have repeated the data shown in different table in the manuscript as well while explanation, I think this is not comfortable and does not make any sense for the repetition of data. This needs to be removed. Authors should use values from different tables to explain their observations and restricts themselves by writing all the number in manuscript.

Answer.

The Table 1, Table 2, Table 3, Table 4, Table 5 and Table 6 were deleted in Manuscript. And the Table 1, Table 2, Table 3, Table 4, Table 5 and Table 6 were changed to Table S1, Table S2, Table S3, Table S4, Table S5 and Table S6 in Supporting Information to help readers understand the degradation performance of NIF via UV/H2O2 system.

Question 29 Authors should perform a study on the effect of NIF removal for different initial concentration of the NIF.

Answer.

The removal of NIF via UV/H2O2 was fast. When the NIF concentration was low, it is difficult to test degradation performance in less than 1 min; however, when the NIF concentration was low, it is not existing in the real water environment. Hence, after many experiments, the NIF concentration = 5 mg/L was selected in this study.

Question 30 Authors have mentioned that pH of the water shows linear decrease with increase in pH value, however as far as I can see the from 4-7 the removal rate is almost constant. How to justify this?

Answer.

The effect of pH was shown in [3.2 Effect of initial pH] as following: 

For [3.2 Effect of initial pH]

The pH is another key parameter of the UV/H2O2 system. It significantly affects the oxidative degradation of antibiotics by transforming the protonation states and changing the redox potential at different pH values. As shown in Fig. 4(a-b) and Table S2, the degradation rates of NIF under UV/H2O2 was 97.54% (pH = 10) < 98.69% (pH = 8) < 99.77% (pH = 5) < 99.94% (pH = 4 and 7), the kinetics constant k’app was 0.75217 min-1 (pH = 10) < 0.91269 min-1 (pH = 8) < 1.21831 min-1 (pH = 5) < 1.45569 min-1 (pH =7) < 1.51175 min-1 (pH = 4), and the t1/2 was 0.4 min (pH =7) < 0.7 min (pH = 4) < 0.8 min (pH = 5) < 0.9 min (pH = 8) < 1.1 min (pH = 10) when the NIF concentration = 5 mg/L, pH = 4-10, H2O2 dose = 0.52 mmol/L, T = 20 ℃ and reaction time = 5 min. As shown in Fig. 4c, the degradation kinetics of NIF via UV/H2O2 system exhibited a poor linear decrease as the pH increased (y = -0.1155x + 1.95533, R2 = 0.6884). The results indicated that acidic solution (pH = 4) was more favorable for degrading NIF than basic solution (pH = 10) under UV/H2O2. The possible reasons were in accord with the redox potential, generation rate of ROS and reaction rate between ROS and NIF. The inhibiting effect of the basic solution was due to the quenching reaction between OH- and •OH, which is shown in Equations 7 (E7) to 11 (E11)[36]:

H2O2 → HO2- + H+ (E7)

H+ + OH- → H2O (E8)

H2O2 + HO2- → H2O + OH- + O2 (E9)

•OH + HO2- → OH- + •HO2 (E10)

•OH + HO2- → H2O + •O2- (E11)

Fig. 4 Effect of initial pH on the removal rate (a), kinetics constant (b) and linear fitting between kinetics constant and initial pH on the degradation of NIF via UV/H2O2. The error bars represent the standard deviation (n = 3).

---

## [Decision Letter · Decision Letter 1]

22 Sep 2021

PONE-D-21-20580R1Highly efficient UV/H2O2 technology for the removal of nifedipine antibiotics: Kinetics, co-existing anions and degradation pathwaysPLOS ONE

Dear Dr. Wang,

Thank you for submitting your manuscript to PLOS ONE. After careful consideration, we feel that it has merit but does not fully meet PLOS ONE’s publication criteria as it currently stands. Therefore, we invite you to submit a revised version of the manuscript that addresses the points raised during the review process.

We look forward to receiving your revised manuscript.

Kind regards,

Van-Huy Nguyen, Ph.D.

Academic Editor

PLOS ONE

Journal Requirements:

Reviewers' comments:

Reviewer's Responses to Questions

**Comments to the Author**

1. If the authors have adequately addressed your comments raised in a previous round of review and you feel that this manuscript is now acceptable for publication, you may indicate that here to bypass the “Comments to the Author” section, enter your conflict of interest statement in the “Confidential to Editor” section, and submit your "Accept" recommendation.

Reviewer #1: All comments have been addressed

Reviewer #2: All comments have been addressed

Reviewer #3: All comments have been addressed

2. Is the manuscript technically sound, and do the data support the conclusions?

Reviewer #1: Yes

Reviewer #2: Yes

Reviewer #3: Yes

3. Has the statistical analysis been performed appropriately and rigorously? 

Reviewer #1: Yes

Reviewer #2: Yes

Reviewer #3: N/A

4. Have the authors made all data underlying the findings in their manuscript fully available?

Reviewer #1: Yes

Reviewer #2: Yes

Reviewer #3: Yes

5. Is the manuscript presented in an intelligible fashion and written in standard English?

Reviewer #1: Yes

Reviewer #2: Yes

Reviewer #3: Yes

6. Review Comments to the Author

Reviewer #1: The revised manuscript can be accepted in the present form. Authors considered the review comments and incorporated in the revised manuscript.

Reviewer #2: (No Response)

Reviewer #3: The novel of study should be mentioned clearly. The combination of UV light with hydrogen peroxide (UV/H2O2) to degrade organic pollutants is not new, so the author should reveal what difference is?

7. PLOS authors have the option to publish the peer review history of their article (what does this mean?). If published, this will include your full peer review and any attached files.

Reviewer #1: **Yes: **Rangabhashiyam S

Reviewer #2: **Yes: **Pardeep Singh

Reviewer #3: No

---

## [Author Response · Author response to Decision Letter 1]

24 Sep 2021

Response to Editor's comments:

Thank you very much for your consideration of our manuscript entitled “Highly efficient UV/H2O2 technology for the removal of nifedipine antibiotics: Kinetics, co-existing anions and degradation pathways” (PONE-D-21-20580R1) to Plos One.

We have read the editor’s comments and reviewers’ questions carefully, and made an attempt to respond as clearly as possible. In reference to the above recommendations, the responses are shown in “Response to 3rd reviewer”.

Response to Journal Requirements:

Answer.

The references were checked without any questions.

The references [15] and [16] were not the retraction paper.

The authors, title, journal, year and doi of references [15] and [16] were correct and searchable although without issue, volume and pages.

[15] Gunarathne V, Rajapaksha AU, Vithanage M, Alessi DS, Selvasembian R, Naushad M, You SM, Oleszczuk P, Ok YS. Hydrometallurgical processes for heavy metals recovery from industrial sludges. Crit Rev Env Sci Tec. 2020; 10.1080/10643389.2020.1847949.

[16] Hariharan A, Harini1 V, Sandhya S, Rangabhashiyam S. Waste Musa acuminata residue as a potential biosorbent for the removal of hexavalent chromium from synthetic wastewater. Biomass Convers Bior. 2020; 10.1007/s13399-020-01173-3.

Response to 3rd reviewer:

Question 12: The novel of study should be mentioned clearly. The combination of UV light with hydrogen peroxide (UV/H2O2) to degrade organic pollutants is not new, so the author should reveal what difference is?

Answer.

Thanks a lot. The novelty of this paper was mentioned in [Introduction] as following:

A novel method of UV light combined with hydrogen peroxide (UV/H2O2) is highly efficient, fast, and has a strong oxidizing ability; these advantages are attributed to the synergistic ability of UV light and H2O2 to generate ROS[30]. However, in UV/H2O2 AOPs, other constituents in water matrices may significantly affect the removal of target contaminants by competitively interacting with photons and ROS. In our previous study, the degradation of norfloxacin by using UV/H2O2 was investigated[31]. The degradation rate and apparent first-order kinetics constant of norfloxacin via UV/H2O2 were 98.8% and 0.22248 min-1, respectively, and the norfloxacin concentration = 20 mg/L, the H2O2 dose =1.2 mmol/L, the pH=7, T= 20 °C and the reaction time = 20 min. The kinetics were low, and the formation mechanism of ROS was controversial, but it provided a novel research direction for the degradation of NIF via a UV/H2O2 system. Therefore, it should be noted that the degree of research to date on the degradation of NIF via UV/H2O2 oxidation processes is insufficient to thoroughly understand the fundamentals of •OH generation, intermediate products and degradation pathways, which are important processes that must be considered in the design of wastewater treatment technology[32]. Furthermore, the effect of co-existing anions in the UV/H2O2 system may significantly affect the removal of NIF by competitively quenching with ROS[33]. Thus, it is still challenging to design a UV/H2O2 wastewater treatment technology with high efficiency.

On the one hand, the oxidizability of UV/H2O2 AOPs and removal rate of NIF were enhanced due to the combination between UV and H2O2[30]. On the other hand, the anion (such as NO3-) was generated due to the degradation reaction between NIF and ROS. However, impacts of NO3- showed duality: it promotes the generation of ROS under irradiation, and also quenches the ROS of UV/H2O2[32-33]. Hence, it is significant and meaningful to study the effect of co-existing anions on the degradation of NIF via UV/H2O2 AOPs. To better understand the removal efficacy of a target compound by UV/H2O2 AOPs in different real water environments, the divalent anions (SO42- and CO32-) and monovalent anions (Cl- and NO3-) have been developed to model the impact of water constituents on the reaction kinetics.

---

## [Editor Report · Decision Letter 2]

29 Sep 2021

Highly efficient UV/H2O2 technology for the removal of nifedipine antibiotics: Kinetics, co-existing anions and degradation pathways

PONE-D-21-20580R2

Dear Dr. Wang,

We’re pleased to inform you that your manuscript has been judged scientifically suitable for publication and will be formally accepted for publication once it meets all outstanding technical requirements.

Kind regards,

Van-Huy Nguyen, Ph.D.

Academic Editor

PLOS ONE
---

## [Editor Report · Acceptance letter]

1 Oct 2021

PONE-D-21-20580R2 

Highly efficient UV/H2O2 technology for the removal of nifedipine antibiotics: Kinetics, co-existing anions and degradation pathways 

Dear Dr. Wang:

I'm pleased to inform you that your manuscript has been deemed suitable for publication in PLOS ONE. Congratulations! Your manuscript is now with our production department. 

Kind regards, 

on behalf of

Dr. Van-Huy Nguyen 

Academic Editor

PLOS ONE